# Comparative Assessment of the Efficacy of the Five Kinds of Models in Landslide Susceptibility Map for Factor Screening: A Case Study at Zigui-Badong in the Three Gorges Reservoir Area, China

**Xianyu Yu** [1,2,*] **, Tingting Xiong** [1] **, Weiwei Jiang** [1,2] **and Jianguo Zhou** [1,2]

1   School of Civil Engineering, Architecture and Environment, Hubei University of Technology,
    Wuhan 430068, China
2   Innovation Demonstration Base of Ecological Environment Geotechnical and Ecological Restoration of Rivers
    and Lakes, Hubei University of Technology, Wuhan 430068, China
*   Correspondence: yuxianyu@hbut.edu.cn

**Abstract:** Landslides are geological disasters affected by a variety of factors that have the characteristics of a strong destructive nature and rapid development and cause major harm to the safety of people's lives and property within the scope of the disaster. Excessive landslide susceptibility mapping (LSM) factors can reduce the accuracy of LSM results and are not conducive to researchers finding the key LSM factors. In this study, with the Three Gorges Reservoir area to the Padang section as an example, the frequency ratio (FR), index of entropy (IOE), Relief-F algorithm, and weights-of-evidence (WOE) Bayesian model were used to sort and screen the importance of 20 LSM factors; then, the LSMs generated based on different factor sets modeled are evaluated and further scored. The results showed that the IOE screening factor was better than the FR, Relief-F, and WOE Bayesian models in the case of retaining no fewer than eight factors; the score for 20 factors without screening was 45 points, and the score for 12 factors screened based on the IOE was 44.8 points, indicating that there was an optimal retention number that had little effect on the LSM results when IOE screening was used. The core factor set obtained by the method for comparing the increase in scores and the increase in corresponding factors effectively improved the accuracy of the LSM results, thus verifying the effectiveness of the proposed method for ranking the importance of LSM factors. The method proposed in this study can effectively screen the key LSM factors and improve the accuracy and scientific soundness of LSM results.

**Keywords:** landslide susceptibility mapping; factor screening; frequency ratio; index of entropy; Relief-F algorithm; weights-of-evidence (WOE) Bayesian model

## 1. Introduction

Based on the frequent occurrence of landslides in the Three Gorges Reservoir area of China, posing a huge risk to people's lives and property in the reservoir area, the efficient and accurate generation of landslide vulnerability maps is of great significance for alleviating the issues caused by landslides [1]. Landslides are harmful geological disasters caused by rapid changes in the natural environment, resulting from the stress applied on the rock and soil exceeding the strength of the soil; thus, the soil along the slope (a slippery surface) moves. This downslope soil movement is characterized by strong destruction, rapid development, and other properties. Geomorphologists, geologists, engineering geologists, and other researchers have combined different strategies and methods to explore landslides in long-term research involving the international community [2]. The occurrence of landslides is influenced by many factors. There are many methods for landslide susceptibility mapping (LSM), but they all use different models to model different factors, and there are few studies on the selection of LSM factors. In order to find a suitable factor-screening method

for landslide susceptibility evaluation, this paper tries to find several classical statistical methods to screen factors and compare them.

LSM, in which factors from engineering geology are used to predict the probability of future landslides by statistically analyzing the factors that lead to landslides, is common in regional landslide studies [3]. In recent years, machine learning models have been widely used in LSM models, and they include logistic regression (LR) [4–8], random forest (RF) [8–11], support vector machine (SVM) [12–15], and artificial neural network (ANN) [16–18] models, among others. There is no single or specific model that can be described as the best scenario for all situations, and the various modeling approaches are compared in this overview. LR models can produce results very quickly and easily, but the conditions of use are strict; the RF model avoids overfitting and outliers, but takes quite a long time to run; SVM can balance overall performance and computation time, but they are also sensitive to data structures; ANN has faster convergence speed and better performance, but the results are difficult to interpret, and a large number of samples are needed to obtain a reliable model [19]. These methods quantify the influence of each factor on the incidence of landslides by identifying the important factors at the time of a landslide and then calculating the values or relative weights of these factors [20]. Therefore, no matter which method is used in LSM, the choice of LSM factor is very important.

Landslides are affected by a variety of factors, including geology, soil, forest cover, topography, and climatic conditions [20–24], where geology, soil, forest cover, and terrain are important internal factors, and foundations for the development of landslides and climate, earthquakes, and human engineering activities are important factors that induce landslides. The extraction of remote sensing images with different resolutions can obtain LSM factors with different classification accuracy, thus affecting the overall accuracy and accuracy of LSM [25]. A large amount of remote sensing data about the Three Gorges Reservoir area can be obtained by using unmanned airborne systems (UASs) and lidar. However, the challenge faced by UASs in data collection is that the increase in data volume leads to the lack of appropriate analytical methods [26]. When analyzing the data, the steep and rugged terrain and severe vegetation cover in the Three Gorges area often cover up the geomorphic features that predict landslides, and the remote sensing image resolution obtained from the original data is too low [25]. Therefore, this study finally decided to abandon the correlation factors extracted from remote sensing images. Due to the differences in geographical locations of landslides, the factors that lead to the occurrence of landslide events are uncertain; therefore, the effects of various factors should be considered in LSM [27]. However, in LSM, the number of factors considered is not always proportional to the effects of a landslide [28], and in fact, only a few factors or combinations of factors contribute significantly to the occurrence of landslides; identifying these factors is critical for accurate LSM [27]. If all LSM factors are considered, not only do correlations and multicollinearity among factors affect the weights during computations, but also considering too many factors can result in a high computational burden and is not conducive to highlighting the results of the study factors when modeling landslide susceptibility [29]. Screening the LSM factors, removing redundancy and collinearity among factors, and reducing the dimensionality of factors are important for strengthening the stability of LSM models [30]. Therefore, the use of appropriate methods to screen the LSM factors is an important research topic.

Statistical methods are the most widely used methods for factor screening in LSM [31]. The principle is to use statistical methods to determine the importance of each factor in relation to the occurrence of landslides and only retain the factors with the greatest contributions. The commonly used statistical methods are the certainty factor (CF) [32–35], frequency ratio (FR) [36–39], index of entropy (IOE) [40,41], weights-of-evidence (WOE) [36,40,42,43], and relief algorithm methods [44–47]. Wu et al. used rough set and correlation coefficient analysis to screen 12 key environmental factors from 22 overall landslide factors for LSM [1]. Dou et al. used the CF method to optimize 14 possible LSM factors and selected slope angle, aspect, diversion density network, dis-

tance from a geological boundary, distance from a fault, and lithology as factors for further analysis [48]. Niu et al. used CF, the sensitivity index (SI), and the correlation coefficient (CC) to analyze the suitability of including nine environmental factors in landslide analysis and finally selected four terrain factors, namely slope, slope of slope, slope shape, and surface roughness, to map landslide susceptibility together with other factors, and the results exhibited reasonable accuracy [49]. Djukem et al. analyzed the geomechanical properties of 11 representative soil samples using the IOE method, calculated the weight values of geological factors, and concluded that the spatial distribution of soil mechanical properties played an important role in the occurrence of landslides [50]. Wang et al. analyzed nine LSM factors using the maximum entropy model, and the results showed that road distance, rainfall, and land use were the main risk factors affecting landslide occurrence [51]. In the above LSM-related research, the screening of factors occurred only as a one-step process, and few papers have specifically studied the screening of landslide factors. There are many methods of screening factors that, to a certain extent, can indeed remove some unnecessary factors and factors that have little impact on LSM, but it is unclear whether the results obtained by using different methods to screen factors are the same and which screening method is most effective.

In this study, the Zigui to Badong section of the Three Gorges reservoir was taken as the research area. 40 LSM factors were obtained by collating and analyzing the collected geological, topographic, and hydrological data, remote sensing images, and other original data. Then, using the Pearson correlation coefficient (PCC) and the variance inflation factor (VIF) to remove redundant, highly correlated, and multicollinear factors, 20 LSM factors were retained. To further screen the LSM factors, the importance of these factors was sorted with four methods—the FR, IOE, Relief-F, and WOE Bayesian models—and different sets of important factors (six factors, eight factors, ten factors and twelve factors) were selected in order of importance from highest to lowest. Then, the same batches of training and validation samples were used to generate LSMs using an SVM. To compare the effectiveness of the four screening methods, the LSM results were evaluated based on the receiver operating characteristic (ROC) curve and the area under the curve (AUC), and the specific category precision analysis and the five statistical measures were applied. The evaluation results were comprehensively and quantitatively scored by using the scoring system for comparisons. Furthermore, to study the importance of LSM factors, the importance of LSM factors was reordered by comparing the score increases and the corresponding factor increases.

## 2. Overview of the Study Area and Data Introduction

### 2.1. Overview of the Research Area

This study area is located in the first area of the Three Gorges Reservoir of the Yangtze River, from Zigui to Badong, covering two county-level administrative districts and eleven township-level administrative districts. The geographical coordinates of the study area are 110°18′~110°52′ east longitude and 30°01′~30°56′ north latitude. The water system in the study area is mainly the Yangtze River and its main tributaries flowing through Padang and Zigui, and the total length of the major river basin is approximately 55 km. In terms of topography, the study area is located in the eastern part of two natural geographical units in the Three Gorges Reservoir area, which is a basin, and the terrain along the river is characterized by low on middle position and high areas on both sides [52]. The Three Gorges area of the Yangtze River was affected by the Quaternary uplift, and lumpy Paleozoic and Mesozoic (Triassic Jialing River Group) limestone is severely cut along the narrow fault zone; the main geological structures in the east-west direction are the Huangning anticline, Zigui oblique, and Padang oblique, and the lithology is composed of Cretaceous rocks, Jurassic rocks, Quaternary loose rocks, Devonian clastic rocks, Triassic and Sinian carbonate rocks, and Precambrian crystalline rocks [53]. Geological disasters occur frequently in the study area, and landslides are the most prominent type of geological disaster. There have been 202 verified landslides with an area of about 23.4 km$^2$, accounting

for 6.03% of the entire study area [54]. A schematic diagram of the location of the study area is shown in Figure 1.

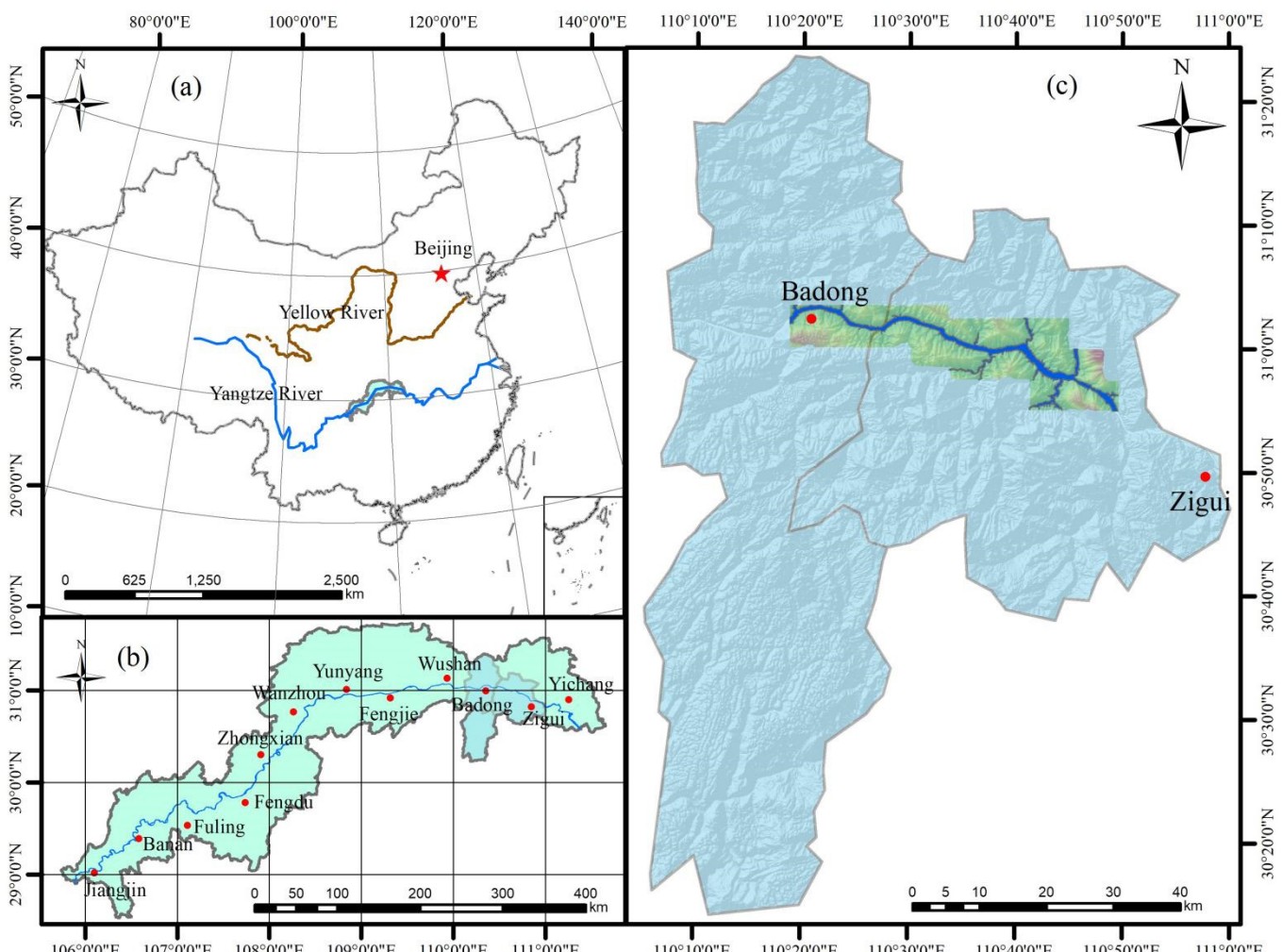

**Figure 1.** Schematic diagram of the location of the study area: (**a**) Map of China; (**b**) Map of Three Gorges Reservoir area; (**c**) Geographical location of the study area.

### 2.2. Data and Software Sources

The data sources used in this study are shown in Table 1.

**Table 1.** Data sources used in this study.

| Name | Data Source | Spatial Resolution/Scale |
|---|---|---|
| DEM data | https://lpdaac.usgs.gov/tools/data~pool/ (accessed on 16 November 2021) | 30 m |
| Basic geographic data | Hubei Geological Survey Institute (accessed on 10 November 2021) | 1:50,000 |
| The landslides distribution data | Landslide hazard map (accessed on 12 November 2021) | 1:10,000 |

The spatial resolution of the DEM data can be matched with the selected topographic map and geological map at a 1:50,000 scale and the landslide disaster map at a 1:10,000 scale. The software used in this study are as follows: ArcGIS 10.8, ENVI 5.3, SPSS Modeler 18, SPSS Statistics 26, and PyTorch 1.7.1. ArcGIS 10.8 and ENVI 5.3 are software developed by ESRI in RedLands, CA, USA; SPSS Modeler 18 and SPSS Statistics 26 are software

developed by IBM in Armonk, New York, USA; PyTorch 1.7.1 is a deep learning framework developed by Facebook in Menlo Park, CA, USA.

### 2.3. Definition of Data

According to the data sources in Table 1, a total of 40 LSM factors were calculated, sorted, and divided into three categories, as shown in Table 2.

**Table 2.** Name, description, and classification of variables.

| Type | Variable | Unit | Value Range |
|------|----------|------|-------------|
| Geology | Fault | m | 0~8719.7 |
| | Lithology | | 1. Hard Rock; 2. Soft–Hard Alternating Rock; 3. Soft Rock. |
| | Slope Structure | | 1. Over-dip Slope; 2. Under-dip Slope; 3. Dip-oblique Slope; 4. Anaclinal Slope; 5. Anaclinal-oblique Slope; 6. Transverse Slope. |
| Terrain | TCI Low | | 0.169229~0.970989 |
| | Terrain Surface Texture | | 0~0.694006 |
| | Total Curvature | | 0~0.118286 |
| | TPI | | −83.3541~227.751 |
| | TRI | | 0~192.657 |
| | Valley Depth | m | 0.00256774~1308.15 |
| | Tangential Curvature | | −0.0749825~0.0468265 |
| | Slope Length | m | 0~3909.74 |
| | Slope Height | m | 0~1227.48 |
| | Slope Form | | 1. Outside Convex Slope; 2. Outside Concave Slope; 3. Outside Straight Slope; 4. Inside Convex Slope; 5. Inside Concave Slope; 6. Inside Straight Slope; 7. Straight Convex Slope; 8. Straight Concave Slope; 9. Straight Slope. |
| | Slope | m | 0.00305845~78.419 |
| | Profile Curvature | | −0.0689291~0.0629155 |
| | Plan Curvature | | −1.96555~4.0909 |
| | Minimal Curvature | | −0.405303~0.0755154 |
| | Mid-slope Position | | 0~1 |
| | Maximal Curvature | | −0.0707668~0.13735 |
| | Longitudinal Curvature | | −0.639569~0.242719 |
| | General Curvature | | −0.652313~0.42573 |
| | Landforms | | 1. Canyons, deeply incised streams; 2. Mid-slope drainages, shallow valleys; 3. Upland drainages, headwaters; 4. U-shape valleys; 5. Plains; 6. Open slopes; 7. Upper slopes, mesas; 8. Local ridges/hills in valleys; 9. Mid-slope ridges, small hills in plains; 10. Mountain tops, high ridges. |
| | Elevation | m | 80~2000 |
| | Cross-sectional Curvature | | −0.243084~0.183011 |
| | Convexity | | 0.170068~0.81388 |
| | Convergence Index | | −74.3747~82.5724 |
| | Aspect | | 1. North; 2. Northeast; 3. East; 4. Southeast; 5. South; 6. Southwest; 7. West; 8. Northwest. |
| Hydrology | Distance from River | m | 0~2395.07 |
| | TWI | | 4.44223~18.03 |
| | VDCN | m | −464.027~1724.15 |
| | SPI | | 0~1136150 |
| | River Buffer Zone | m$^2$ | 377.319~4559.85 |
| | MRN | | 0~37.728 |
| | LS Factor | | 0~95.7068 |
| | Flow Line Curvature | | −2.72377~0.346225 |
| | Flow Path Length | m | 0~34.8072 |
| | Flow Width | m | 28.5~40.3051 |
| | CNBL | | 80.2274~1353.9 |
| | Catchment Slope | | 0~1.46657 |
| | Catchment Area | m$^2$ | 812.25~1637870 |

### 2.4. Create Training Samples and Verify the Samples

According to Table 1, 30 m × 30 m grid cells are used for calculation in this research area. After removing invalid data, 423,787 raster cells were obtained. Among them, there were 25,213 landslide grid units and 398,574 non-landslide grid units. In the study, 70% of the landslide grid units and the same number of non-landslide grid units were randomly

selected as training samples, and the remainder of the grid units were used as validation samples. The SVM was used for LSM.

## 3. Introduction to the Research Methods

### *3.1. Factor Pretreatment Method*

#### 3.1.1. PCC

The Pearson correlation coefficient, also known as the Pearson moment correlation coefficient, is a linear correlation coefficient that reflects the degree of linear correlation between two variables [55]. PCC is the degree of linear correlation between two variables, where a greater absolute value indicates a stronger correlation. The value of the PCC is between $-1$ and $1$, and the correlation between two variables can vary from a negative correlation to a positive correlation. When the PCC is 0, the two variables have no correlation, which means that the two variables are independent of each other [56]. This article treats two variables with PCC greater than 0.6 as having a strong correlation, and one of the variables should be removed to eliminate correlation issues during analyses.

#### 3.1.2. VIF

VIF is a measure of the severity of multicollinearity in multiple linear regression models [57]. It represents the ratio of the variance of the regression coefficient estimate to the variance of the hypothesized nonlinear correlation between two independent variables. In this article, the variables whose application range of VIF is 1–10 are regarded as reasonable factors of multicollinearity.

### *3.2. Factor-Screening Method*

#### 3.2.1. FR

The correlation between the landslide distribution and landslide genesis can be deduced using FR, defined by the ratio of the region where landslides occur to the entire study area and the ratio of landslide occurrence probabilities to nonoccurrence probabilities considering a given property [4].

#### 3.2.2. IOE

The fuzzy comprehensive evaluation method is a comprehensive evaluation method based on fuzzy theory, and it is used to describe the uncertainty of things; additionally, the degree of dispersion of an index can be assessed based on the entropy value of information [58]. The greater the degree of dispersion of an indicator, the smaller the weight, and vice versa.

#### 3.2.3. Relief-F Algorithm

As a separate method of evaluating filtered feature selection, Relief calculates an agent statistic for each feature, and these results can be used to estimate feature quality or correlations with a target process (i.e., predicting endpoint values). The original Relief algorithm has been rarely applied in practical applications and has been replaced by Relief-F [59]. The principle of the Relief-F algorithm is to randomly extract a sample R from the sample set T; then, k neighboring samples (H) of R are selected from the sample set of R, and k neighboring samples (N) of R are selected from the sample set of different classes of R; the whole process is repeated m times [60].

When using the Relief-F algorithm to evaluate the predictive ability of different landslide evaluation factors, a large value indicates that the weight of the evaluation factor should be high, which means that the corresponding feature influence is strong; otherwise, it indicates that the weight of the evaluation factor is lower [61].

#### 3.2.4. WOE Bayesian Model

The WOE Bayesian model combines weights-of-evidence and a Bayesian formula. The loglinear form of the Bayesian probability model is used to predict the WOE of each

independent variable in the independent variable group of dependent variables and the importance of discriminating independent variables; it is a standard, quantitative, data-driven statistical method [62].

### 3.3. SVM

The SVM model, first proposed by Vapnik [63], has many unique advantages for solving small-sample, nonlinear, and high-dimensional pattern recognition problems and can be generalized to other machine learning problems, such as function fitting [7,64]. Assuming that a linearly separable training vector $\chi_i (i = 1, 2, \ldots \ldots, n)$ contains two classes $y_i = \pm 1$, an n−1-dimensional hyperplane needs to be searched so that the two classes are separate and spaced as far apart as possible in the SVM [65].

### 3.4. Evaluation Methods

To clarify the influence of different combinations of factors obtained with different screening methods on the LSM results, the different sets of LSM results were analyzed from different perspectives by using the ROC curve, the specific category precision analysis method, and the five statistical measures.

#### 3.4.1. ROC Curve Analysis

The ROC curve is a useful method for assessing the predictive power of a model, and it is often used in LSM. The longitudinal axis of the ROC curve is the true positive rate (TPR = TP/(TP + FN)), the horizontal axis is the false positive rate (FPR = FP/(FP + TN)), and the ROC curve starts at point (0,0) and reaches (1,1) [66–68]. This approach is detailed in Table 3.

**Table 3.** Confusion matrix.

| Confusion Matrix | | Predicted Value | |
|---|---|---|---|
| | | **Positive** | **Negative** |
| **Observed Value** | Positive | True Positive, TP | False Negative, FN |
| | Negative | False Positive, FP | True Negative, TN |

#### 3.4.2. Specific Category Precision Analysis

The specific category precision analysis method is an improved quantitative analysis method that takes into account the number of computational units within the prediction area [52]. This method can be expressed as:

$$P_i = \frac{A_i}{B_i} \times 100\% \tag{1}$$

where $i$ = 1,2 . . . , n; n is the number of landslide zoning classifications; $A_i$ is the number of slope units occupied by landslides in the $i$th landslide-prone area; $B_i$ is the number of slope units in the $i$th landslide-prone zone; and $P_i$ is the specific classification accuracy of the division of the $i$th landslide-prone area.

#### 3.4.3. Evaluation with Five Statistical Measures

Overall accuracy (OA), precision, recall, the F-measure, and the Matthews correlation coefficient (MCC) are five common statistical measures used to assess the ability of classification models based on confusion matrices [69,70]. In LSM, the higher the OA value, the higher the prediction accuracy of the whole study area. The higher the precision value, the higher the prediction accuracy of a landslide. The higher the recall value, the higher the proportion of landslides correctly predicted in actual landslides. The F-measure is the weighted harmonized average of precision and recall, and a high F-measure indicates that the test method is effective. The MCC describes the correlation between the actual

classification and the predicted classification; a value of 1 indicates a perfect prediction, a value of 0 indicates that the predicted result is not as good as a randomly predicted result, a value of $-1$ indicates that the predicted classification and the actual classification are completely inconsistent.

## 4. Experiments

The overall workflow of this study is shown in Figure 2.

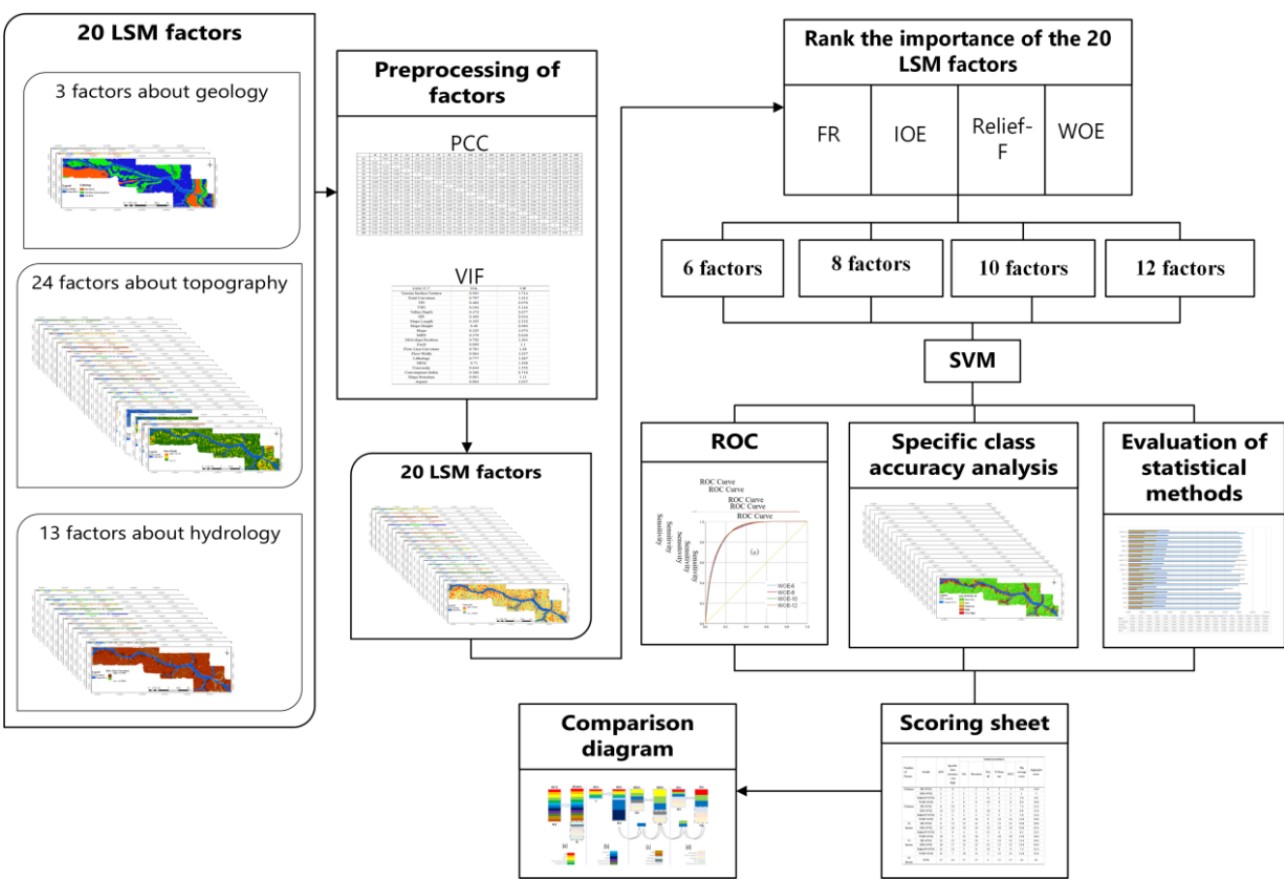

**Figure 2.** Overall workflow of this study.

### 4.1. Data Preprocessing

To reduce the collinearity and redundancy of landslide-related factors, the PCC was used to generate a correlation matrix. Then, the strongly correlated factors (correlation greater than 0.6) were removed, and 20 LSM factors (terrain surface texture, total curvature, TPI, TWI, valley depth, SPI, slope length, slope height, slope, MRN, mid-slope position, fault, flow line curvature, flow width, lithology, elevation, convexity, convergence index, slope structure, and aspect) were screened. The results are shown in Figure 3.

|     | A1 | A2 | A3 | A4 | A5 | A6 | A7 | A8 | A9 | A10 | A11 | A12 | A13 | A14 | A15 | A16 | A17 | A18 | A19 | A20 |
|-----|----|----|----|----|----|----|----|----|----|-----|-----|-----|-----|-----|-----|-----|-----|-----|-----|-----|
| A1 | 1 | 0.055 | 0.01 | -0.183 | -0.37 | -0.101 | -0.288 | -0.312 | -0.175 | -0.35 | 0.129 | -0.052 | -0.001 | 0.019 | 0.107 | 0.158 | 0.002 | 0.033 | 0.145 | 0.022 |
| A2 |  | 1 | -0.115 | -0.05 | 0.054 | 0.136 | 0.016 | 0.011 | 0.205 | 0.048 | 0.073 | -0.02 | -0.45 | 0.007 | -0.036 | 0.031 | -0.013 | -0.037 | 0.054 | 0.001 |
| A3 |  |  | 1 | -0.561 | -0.368 | -0.357 | -0.349 | 0.358 | 0.006 | -0.371 | -0.025 | 0 | 0.017 | 0.011 | -0.013 | 0.114 | 0.135 | 0.48 | 0.052 | -0.006 |
| A4 |  |  |  | 1 | 0.553 | 0.504 | 0.553 | -0.255 | -0.446 | 0.195 | 0.139 | -0.053 | 0.006 | -0.017 | -0.025 | -0.197 | -0.208 | -0.496 | -0.144 | -0.019 |
| A5 |  |  |  |  | 1 | 0.522 | 0.542 | -0.077 | 0.165 | 0.489 | 0.06 | -0.011 | 0 | -0.039 | -0.135 | -0.178 | -0.28 | -0.484 | -0.117 | 0.039 |
| A6 |  |  |  |  |  | 1 | 0.512 | -0.121 | 0.051 | 0.168 | 0.179 | 0.004 | -0.002 | -0.002 | -0.04 | -0.074 | -0.105 | -0.286 | -0.054 | 0.016 |
| A7 |  |  |  |  |  |  | 1 | -0.114 | 0.007 | 0.57 | 0.021 | -0.013 | 0.001 | -0.049 | -0.064 | -0.14 | -0.152 | -0.354 | -0.116 | -0.012 |
| A8 |  |  |  |  |  |  |  | 1 | 0.106 | -0.053 | 0.146 | -0.022 | -0.002 | -0.009 | -0.17 | 0.354 | 0.36 | 0.515 | -0.04 | 0.019 |
| A9 |  |  |  |  |  |  |  |  | 1 | 0.418 | -0.262 | 0.085 | -0.01 | 0.036 | -0.051 | 0.06 | 0.001 | -0.08 | 0.097 | 0.039 |
| A10 |  |  |  |  |  |  |  |  |  | 1 | -0.177 | 0.017 | -0.002 | -0.038 | -0.081 | -0.075 | -0.175 | -0.36 | -0.034 | 0.01 |
| A11 |  |  |  |  |  |  |  |  |  |  | 1 | 0.005 | -0.001 | -0.014 | 0.012 | 0.077 | 0.049 | 0.077 | 0.021 | 0.011 |
| A12 |  |  |  |  |  |  |  |  |  |  |  | 1 | 0.002 | 0.005 | 0.268 | -0.094 | -0.011 | 0.004 | 0.083 | -0.016 |
| A13 |  |  |  |  |  |  |  |  |  |  |  |  | 1 | 0.005 | -0.004 | 0.002 | 0.007 | -0.001 | -0.002 | 0.002 |
| A14 |  |  |  |  |  |  |  |  |  |  |  |  |  | 1 | 0.029 | 0.001 | 0.019 | 0.012 | -0.01 | 0.158 |
| A15 |  |  |  |  |  |  |  |  |  |  |  |  |  |  | 1 | -0.293 | -0.034 | -0.004 | 0.229 | 0.023 |
| A16 |  |  |  |  |  |  |  |  |  |  |  |  |  |  |  | 1 | 0.187 | 0.249 | -0.056 | 0.028 |
| A17 |  |  |  |  |  |  |  |  |  |  |  |  |  |  |  |  | 1 | 0.547 | -0.008 | 0.015 |
| A18 |  |  |  |  |  |  |  |  |  |  |  |  |  |  |  |  |  | 1 | 0.041 | -0.007 |
| A19 |  |  |  |  |  |  |  |  |  |  |  |  |  |  |  |  |  |  | 1 | 0.032 |
| A20 |  |  |  |  |  |  |  |  |  |  |  |  |  |  |  |  |  |  |  | 1 |

**Figure 3.** PCCs of 20 LSM factors. A1: Terrain surface texture; A2: Total curvature; A3: TPI; A4: TWI; A5: Valley depth; A6: SPI; A7: Slope length; A8: Slope height; A9: Slope; A10: MRN; A11: Mid-slope position; A12: Fault; A13: Flow line curvature; A14: Flow width; A15: Lithology; A16: Elevation; A17: Convexity; A18: Convergence index; A19: Slope structure; A20: Aspect.

Multicollinearity analyses were performed by the VIFs for the 20 LSM factors screened above, and the results are shown in Table 4.

**Table 4.** Multicollinearity analyses of landslide susceptibility evaluation factors.

| LSM Factor | TOL | VIF |
|------------|-----|-----|
| Terrain Surface Texture | 0.583 | 1.714 |
| Total Curvature | 0.707 | 1.414 |
| TPI | 0.482 | 2.076 |
| TWI | 0.194 | 5.144 |
| Valley Depth | 0.374 | 2.677 |
| SPI | 0.492 | 2.034 |
| Slope Length | 0.395 | 2.532 |
| Slope Height | 0.48 | 2.084 |
| Slope | 0.325 | 3.074 |
| MRN | 0.379 | 2.636 |
| Mid-slope Position | 0.792 | 1.263 |
| Fault | 0.909 | 1.1 |
| Flow Line Curvature | 0.781 | 1.28 |
| Flow Width | 0.964 | 1.037 |
| Lithology | 0.777 | 1.287 |
| Elevation | 0.71 | 1.408 |
| Convexity | 0.644 | 1.552 |
| Convergence Index | 0.368 | 2.716 |
| Slope Structure | 0.901 | 1.11 |
| Aspect | 0.964 | 1.037 |

All the factors in Table 4 meet the conditions of TOL > 0.1 and VIF < 10, and the 20 factors selected passed the multicollinearity test.

In summary, the initial 20 landslide factors that passed the PCC and VIF tests are shown in Figure 4.

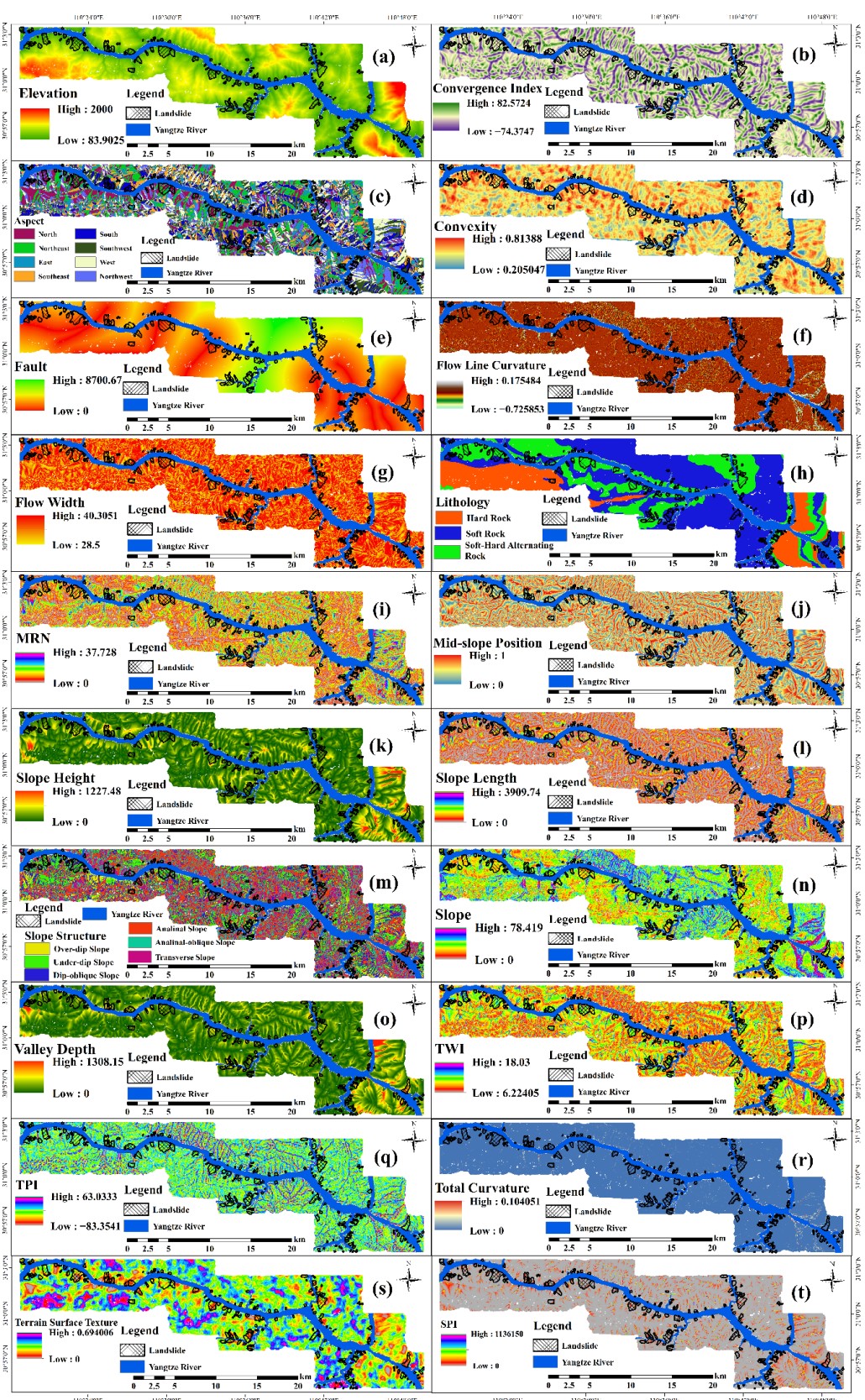

**Figure 4.** LSM factors in the study area: (**a**) Elevation, (**b**) Convergence index, (**c**) Aspect, (**d**) Convexity, (**e**) Fault, (**f**) Flow line curvature, (**g**) Flow width, (**h**) Lithology, (**i**) MRN, (**j**) Mid-slope position, (**k**) Slope height, (**l**) Slope length, (**m**) Slope structure, (**n**) Slope, (**o**) Valley depth, (**p**) TWI, (**q**) TPI, (**r**) Total curvature, (**s**) Terrain surface texture, and (**t**) SPI.

### 4.2. Factor Importance Screening

After 40 factors were screened based on the PCC and VIF tests to obtain 20 LSM factors, the highly correlated and multicollinear factors were eliminated. However, the factors that had little impact on the LSM results were not screened and removed. Therefore, an importance analysis of the landslide factors was performed. In this study, four methods, the FR, IOE, Relief-F, and WOE Bayesian modeling methods, were used to screen the LSM factors.

The relationship between landslide occurrence and the LSM factors was calculated using the FR, IOE, Relief-F algorithm, and WOE Bayesian modeling methods, and the results are shown in Table 5.

**Table 5.** Spatial relationship between each factor and landslide occurrence based on the FR, IOE, Relief-F, and WOE Bayesian modeling methods.

| Factor | Range of Values for Classification | The Size of Study Area | The Size of Landslide | FR | IOE | Relief-F | WOE Bayesian | |
|---|---|---|---|---|---|---|---|---|
| | | | | | | | $W_{final}$ | The AUC without the Factor |
| Terrain Surface Texture | ≤0.1 | 7729 | 1038 | 2.123041519 | 131,882.12 | 3.03% | 25.15672791 | 0.845 |
| | 0.1~0.2 | 48,118 | 7521 | 2.470884598 | | | 84.71821732 | |
| | 0.2~0.3 | 125,372 | 10,258 | 1.293440877 | | | 32.40448146 | |
| | 0.3~0.4 | 131,362 | 5074 | 0.610611743 | | | −43.98146634 | |
| | 0.4~0.5 | 67,532 | 1116 | 0.261239608 | | | −49.86246713 | |
| | >0.5 | 18,461 | 206 | 0.17639913 | | | −26.02038569 | |
| Total Curvature | ≤0.00001 | 95,547 | 11,052 | 1.828555986 | 2459.21 | 7.41% | 73.89226925 | 0.849 |
| | 0.00001~0.0001 | 195,487 | 11,557 | 0.934569649 | | | −10.52490482 | |
| | 0.0001~0.0002 | 47,941 | 1487 | 0.490329837 | | | −30.10794876 | |
| | 0.0002~0.0003 | 20,702 | 504 | 0.38485991 | | | −22.62282999 | |
| | 0.0003~0.0004 | 11,055 | 219 | 0.313162542 | | | −17.89668609 | |
| | 0.0004~0.0005 | 6908 | 123 | 0.281473312 | | | −14.56513525 | |
| | >0.0005 | 20,934 | 271 | 0.204645176 | | | −27.40032779 | |
| TPI | ≤−10 | 33,366 | 889 | 0.421193884 | 140,327.93 | 6.78% | −27.66180398 | 0.848 |
| | −10~−5 | 41,279 | 2232 | 0.854770379 | | | −8.087770482 | |
| | −5~0 | 101,741 | 9347 | 1.452314529 | | | 42.93694928 | |
| | 0~5 | 131,017 | 10,682 | 1.288870739 | | | 32.94845718 | |
| | 5~10 | 57,159 | 1746 | 0.482885382 | | | −33.73766375 | |
| | 10~15 | 22,288 | 271 | 0.192212945 | | | −28.50621446 | |
| | >15 | 11,724 | 46 | 0.062024956 | | | −19.41890095 | |
| TWI | ≤8 | 19,973 | 83 | 0.065693021 | 142,407.35 | 0.23% | −25.73264017 | 0.86 |
| | 8~10 | 264,095 | 9672 | 0.578949323 | | | −92.00555826 | |
| | 10~12 | 94,323 | 13,521 | 2.266082145 | | | 108.1623129 | |
| | 12~14 | 18,043 | 1822 | 1.596335107 | | | 21.05945154 | |
| | >14 | 2140 | 115 | 0.849510025 | | | −1.811694153 | |
| Valley Depth | ≤50 | 148,986 | 3528 | 0.374341139 | 65,898.16 | 4.25% | −74.03853917 | 0.856 |
| | 50~100 | 108,418 | 6039 | 0.880537952 | | | −11.96593041 | |
| | 100~150 | 61,943 | 5447 | 1.390111325 | | | 27.26438873 | |
| | 150~200 | 33,694 | 4086 | 1.917035839 | | | 44.61237656 | |
| | >200 | 45,533 | 6113 | 2.122328333 | | | 63.81352851 | |
| SPI | ≤1000 | 57,270 | 1164 | 0.32129964 | 10,667.48 | 2.93% | −42.68162881 | 0.856 |
| | 1000~4000 | 174,786 | 8952 | 0.809651027 | | | −27.46588053 | |
| | 4000~7000 | 76,428 | 6161 | 1.274333662 | | | 21.83437605 | |
| | 7000~10,000 | 31,463 | 3099 | 1.557061933 | | | 26.46448039 | |
| | >10,000 | 58,627 | 5837 | 1.573897564 | | | 38.5863394 | |
| Slope Length | ≤100 | 194,940 | 8034 | 0.651501331 | 44,684.15 | 3.78% | −54.82417952 | 0.855 |
| | 100~200 | 82,276 | 4447 | 0.854433763 | | | −12.1606934 | |
| | 200~300 | 48,620 | 3437 | 1.117503827 | | | 7.18036755 | |
| | 300~400 | 26,196 | 2408 | 1.453134929 | | | 19.56950679 | |
| | 400~500 | 16,000 | 1806 | 1.784358872 | | | 25.85395212 | |
| | 500~600 | 9640 | 1360 | 2.23021286 | | | 30.7159179 | |
| | >600 | 20,902 | 3721 | 2.814208484 | | | 65.87992865 | |
| Slope Height | ≤20 | 49,193 | 1883 | 0.605105991 | 71,790.71 | 4.30% | −23.98276954 | 0.847 |
| | 20~70 | 136,262 | 9922 | 1.151089003 | | | 17.83047751 | |
| | 70~120 | 87,219 | 7171 | 1.299729753 | | | 25.89550671 | |
| | 120~170 | 50,705 | 3224 | 1.005144932 | | | 0.322254366 | |
| | 170~220 | 28,954 | 1555 | 0.848997213 | | | −6.924069326 | |
| | 220~270 | 16,583 | 711 | 0.677783421 | | | −10.93334835 | |
| | >270 | 29,658 | 747 | 0.398165092 | | | −26.86361557 | |

**Table 5.** *Cont.*

| Factor | Range of Values for Classification | The Size of Study Area | The Size of Landslide | FR | IOE | Relief-F | WOE Bayesian | |
|---|---|---|---|---|---|---|---|---|
| | | | | | | | $W_{final}$ | The AUC without the Factor |
| Slope | ≤10 | 14,971 | 673 | 0.710638439 | 135,956.40 | 2.59% | −9.325273695 | 0.843 |
| | 10~20 | 83,167 | 8374 | 1.591718859 | | | 49.03884418 | |
| | 20~30 | 152,524 | 12,208 | 1.265292039 | | | 34.03781439 | |
| | 30~40 | 105,169 | 3486 | 0.523991304 | | | −45.51085502 | |
| | >40 | 42,743 | 472 | 0.174566715 | | | −40.66036184 | |
| MRN | ≤1 | 76,417 | 3139 | 0.649360359 | 87,202.25 | 4.27% | −27.70607953 | 0.852 |
| | 1~4 | 104,928 | 5976 | 0.900333967 | | | −9.766364922 | |
| | 4~7 | 106,130 | 7280 | 1.084370406 | | | 8.334278476 | |
| | 7~10 | 62,925 | 4897 | 1.230244186 | | | 16.3112819 | |
| | >10 | 48,174 | 3921 | 1.286674149 | | | 17.37371656 | |
| Mid-slope Position | ≤0.1 | 39,380 | 3072 | 1.233189848 | 102,685.71 | 2.44% | 12.63613011 | 0.847 |
| | 0.1~0.3 | 79,611 | 6386 | 1.268061381 | | | 21.88532577 | |
| | 0.3~0.5 | 81,371 | 6253 | 1.214795618 | | | 17.80136222 | |
| | 0.5~0.7 | 84,244 | 5245 | 0.984217209 | | | −1.340464374 | |
| | 0.7~0.9 | 88,206 | 3669 | 0.657557938 | | | −29.62068636 | |
| | >0.9 | 25,762 | 588 | 0.360813012 | | | −26.23025167 | |
| Fault | ≤1500 | 166,254 | 10,637 | 1.011419908 | 105,921.56 | 19.69% | 1.584897163 | 0.848 |
| | 1500~3000 | 109,824 | 6951 | 1.000540038 | | | 0.054640473 | |
| | 3000~4500 | 52,646 | 3123 | 0.937758579 | | | −3.982618149 | |
| | 4500~6000 | 39,685 | 2263 | 0.901452008 | | | −5.373160425 | |
| | 6000~7500 | 25,964 | 2194 | 1.335824683 | | | 14.47859593 | |
| | >7500 | 4201 | 45 | 0.169334041 | | | −12.26434224 | |
| Flow Line Curvature | ≤−0.001 | 42,035 | 833 | 0.3132697 | 42,314.63 | 8.91% | −36.19766115 | 0.846 |
| | −0.001~−0.0005 | 30,210 | 1231 | 0.644157057 | | | −16.55531373 | |
| | −0.0005~0 | 128,052 | 10,662 | 1.316245058 | | | 35.42803841 | |
| | 0~0.0005 | 126,273 | 10,603 | 1.327402723 | | | 36.2891063 | |
| | 0.0005~0.001 | 30,470 | 1160 | 0.601824656 | | | −18.55144031 | |
| | >0.001 | 41,534 | 724 | 0.27556195 | | | −37.38392252 | |
| Flow Width | ≤30 | 34,413 | 2276 | 1.045524381 | 74,675.20 | 0.64% | 2.295696285 | 0.849 |
| | 30~35 | 108,372 | 7317 | 1.067334157 | | | 6.748403248 | |
| | 35~40 | 194,850 | 12,174 | 0.987682431 | | | −1.976271103 | |
| | >40 | 60,939 | 3446 | 0.893931809 | | | −7.387354715 | |
| Lithology | Hard Rock | 78,421 | 802 | 0.161668881 | 38,710.95 | 3.61% | −57.54254831 | 0.836 |
| | Soft-Hard Alternating Rock | 111,696 | 13,085 | 1.851912861 | | | 83.76213169 | |
| | Soft-Rock | 208,457 | 11,326 | 0.858903782 | | | −24.15633662 | |
| Elevation | ≤200 | 27,100 | 6729 | 3.925235146 | 120,609.11 | 7.32% | 115.4239629 | 0.803 |
| | 200~300 | 50,157 | 9401 | 2.962967866 | | | 112.4825669 | |
| | 300~400 | 52,959 | 4986 | 1.488322131 | | | 31.05621469 | |
| | 400~500 | 55,967 | 2891 | 0.816583321 | | | −12.13363048 | |
| | 500~600 | 53,602 | 822 | 0.242423806 | | | −44.34313919 | |
| | 600~700 | 44,764 | 192 | 0.067804229 | | | −39.54329963 | |
| | >700 | 114,025 | 192 | 0.026618623 | | | −55.77986673 | |
| Convexity | ≤0.4 | 16,961 | 1762 | 1.642248566 | 124,943.95 | 0.72% | 21.92797827 | 0.85 |
| | 0.4~0.5 | 110,925 | 8557 | 1.219485206 | | | 22.28710735 | |
| | 0.5~0.6 | 207,790 | 12,236 | 0.930891933 | | | −11.82278933 | |
| | >0.6 | 62,898 | 2658 | 0.668040176 | | | −23.35665684 | |
| Convergence Index | ≤−30 | 36,438 | 1333 | 0.578309144 | 129,519.44 | 6.37% | −21.61042251 | 0.85 |
| | −30~−10 | 81,254 | 6129 | 1.192420168 | | | 15.94036802 | |
| | −10~10 | 129,174 | 12,400 | 1.517508102 | | | 57.66968393 | |
| | 10~30 | 97,150 | 4723 | 0.7685278 | | | −21.4584672 | |
| | >30 | 54,558 | 628 | 0.181964071 | | | −46.4254435 | |
| Slope Structure | Over-dip Slope | 19,196 | 548 | 0.451288492 | 96,820.98 | 3.66% | −19.6470487 | 0.848 |
| | Under-dip Slope | 71,225 | 5792 | 1.285525029 | | | 21.75906997 | |
| | Dip-oblique Slope | 69,169 | 4932 | 1.127187106 | | | 9.552974322 | |
| | Anaclinal Slope | 123,813 | 7630 | 0.974187903 | | | −2.842537665 | |
| | Anaclinal-oblique Slope | 59,350 | 3586 | 0.955155329 | | | −3.076989636 | |
| | Transverse Slope | 55,821 | 2725 | 0.771708592 | | | −15.05732658 | |

**Table 5.** *Cont.*

| Factor | Range of Values for Classification | The Size of Study Area | The Size of Landslide | FR | IOE | Relief-F | WOE Bayesian | |
|---|---|---|---|---|---|---|---|---|
| | | | | | | | W<sub>final</sub> | The AUC without the Factor |
| Aspect | North | 57,060 | 5646 | 1.564204561 | | | 37.35659867 | |
| | Northeast | 55,702 | 4339 | 1.231411776 | | | 15.26219266 | |
| | East | 42,955 | 2443 | 0.899071405 | | | −5.751234837 | |
| | Southeast | 45,540 | 2037 | 0.707102616 | 80,616.50 | 7.06% | −17.14717011 | 0.847 |
| | South | 47,629 | 3493 | 1.159341984 | | | 9.618338101 | |
| | Southwest | 45,425 | 1627 | 0.566209378 | | | −25.09301787 | |
| | West | 53,524 | 1979 | 0.584496175 | | | −26.43084564 | |
| | Northwest | 50,739 | 3649 | 1.136884646 | | | 8.568868387 | |

According to Table 5, the factor importance was sorted by four screening methods from highest to lowest, and the results are shown in Table 6.

**Table 6.** Importance ranking of LSM factors based on different screening methods.

| Importance Ranking | FR | IOE | Relief-F | WOE |
|---|---|---|---|---|
| 1 | Elevation | TWI | Fault | Elevation |
| 2 | Slope Length | TPI | Flow Line Curvature | Lithology |
| 3 | Terrain Surface Texture | Slope | Total Curvature | Slope |
| 4 | TWI | Terrain Surface Texture | Elevation | Terrain Surface Texture |
| 5 | Valley Depth | Convergence Index | Aspect | Flow Line Curvature |
| 6 | Lithology | Convexity | TPI | Slope Height |
| 7 | Total Curvature | Elevation | Convergence Index | Mid-slope Position |
| 8 | Convexity | Fault | Slope Height | Aspect |
| 9 | Slope | Mid-slope Position | MRN | TPI |
| 10 | SPI | Slope Structure | Valley Depth | Fault |
| 11 | Aspect | MRN | Slope Length | Slope Structure |
| 12 | Convergence Index | Aspect | Slope Structure | Total Curvature |
| 13 | TPI | Flow Width | Lithology | Flow Width |
| 14 | Fault | Slope Height | Terrain Surface Texture | Convexity |
| 15 | Flow Line Curvature | Valley Depth | SPI | Convergence Index |
| 16 | Slope Height | Slope Length | Slope | MRN |
| 17 | MRN | Flow Line Curvature | Mid-slope Position | Slope Length |
| 18 | Slope Structure | Lithology | Convexity | Valley Depth |
| 19 | Mid-slope Position | SPI | Flow Width | SPI |
| 20 | Flow Width | Total Curvature | TWI | TWI |

### 4.3. SVM Modeling

According to Table 6, the most important six factors, eight factors, ten factors, and twelve factors in each group were selected as LSM factors, with a total of 16 groups. Plus 20 factors as a group of LSM factors, there were a total of 17 groups of LSM factors. SVM was used to model these 17 groups of factors to generate landslide susceptibility index (LSI). The SVM modeling processes are shown in Figure 5.

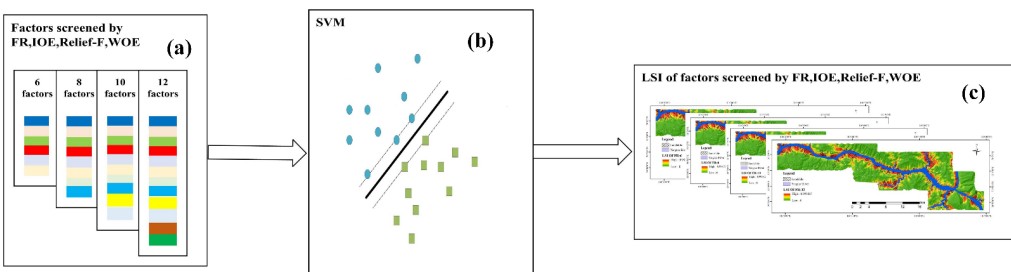

**Figure 5.** Different important factor sets were selected for SVM modeling based on the importance ranking of LSM factors: (**a**) LSM factor sets generated after FR, IOE, Relief-F, WOE screening; (**b**) Modeling with SVM; (**c**) LSI generated by LSM (see Figure 6 for details).

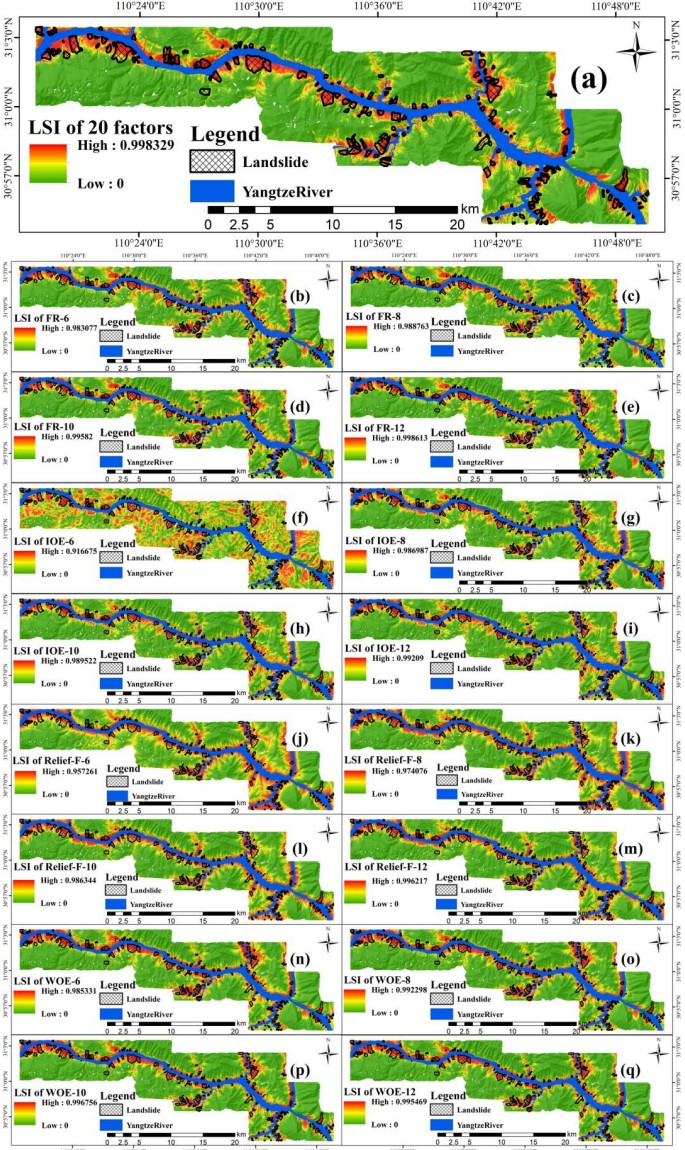

**Figure 6.** LSI obtained with (**a**) 20 factors, (**b**) 6 factors based on FR, (**c**) 8 factors based on FR, (**d**) 10 factors based on FR, (**e**) 12 factors based on FR, (**f**) 6 factors based on IOE, (**g**) 8 factors based on IOE, (**h**) 10 factors based on IOE, (**i**) 12 factors based on IOE, (**j**) 6 factors based on Relief-F, (**k**) 8 factors based on Relief-F, (**l**) 10 factors based on Relief-F, (**m**) 12 factors based on Relief-F, (**n**) 6 factors based on the WOE Bayesian model, (**o**) 8 factors based on the WOE Bayesian model, (**p**) 10 factors based on the WOE Bayesian model, and (**q**) 12 factors based on the WOE Bayesian model.

### 4.4. Experimental Results

4.4.1. LSI Chart

To compare the effectiveness of different screening methods at different screening degrees, four sets of trials were conducted in this study. Based on Table 5, in the first, second, third, and fourth experiments, the top six, eight, ten, and twelve important factors were selected by each screening method. The training samples for the four sets of factors and all the factors were input into the SVM. Then, the LSM model was established, and the LSI of the study area was obtained by using all the samples. The experimental results are shown in Figure 6.

4.4.2. Landslide Susceptibility Zonation (LSZ)

LSZ is a continuous variable that ranges from 0 to 1, and in this study, to increase the readability of landslide vulnerability index graphs, all landslide susceptibility indices were divided into five susceptibility levels using the manual threshold method based on the calculation results: very low (0–0.5), low (0.5–0.7), moderate (0.7–0.8), high (0.8–0.9), and very high (0.9–1.0) [71,72]. This approach is shown in Figure 7.

### 4.5. Analysis of the Experimental Results

4.5.1. ROC Curve

Using SPSS statistical software, the LSM results generated based on the factors screened by the FR, IOE, Relief-F algorithm, and WOE Bayesian modeling methods were used to construct ROC curves. Additionally, AUC values were calculated, and the results are shown in Figure 8 and Table 7.

**Table 7.** AUC values of the four types of models.

| Classifiers | 6 Factors | 8 Factors | 10 Factors | 12 Factors | 20 Factors |
|:---:|:---:|:---:|:---:|:---:|:---:|
| FR | 0.8900 | 0.8945 | 0.9003 | 0.9029 | |
| IOE | 0.8041 | 0.9061 | 0.9064 | 0.9103 | |
| Relief-F | 0.8686 | 0.8908 | 0.8931 | 0.9025 | 0.9107 |
| WOE | 0.8974 | 0.9014 | 0.9023 | 0.9038 | |

In Figure 8, (a) compares the use of different screening methods to the same degree and compares the results based on 20 factors. In Figure 8, (b)–(e) compare the different degrees of screening using the same methods. Figure 8 shows that as the number of factors increases, the area under the ROC curve increases. In plot (a), the ROC curves of FR-12, IOE-12, Relief-F-12, WOE-12, and LSM with 20 factors are similar. As shown in plots (b) and (e), the ROC curves of FR-6, FR-8, FR-10, and FR-12 are very similar, the ROC curves of WOE-6, WOE-8, WOE-10, and WOE-12 are very similar, and the different degrees of factor screening with the FR and WOE Bayesian modeling methods have little effect on the ROC curve in LSM. For IOE screening in plot (c), the ROC curves of IOE-8, IOE-10, and IOE-12 are very similar, but they are very different from the ROC curve of IOE-6. In plot (d), the ROC curves of Relief-F-8, Relief-F-10, and Relief-F-12 are very similar and plot far from the ROC curve of Relief-F-6.

Table 7 shows that the AUC value increases as the number of factors increases. Notably, when the number of factors increased from six to eight, the AUC values changed little, except that of the IOE method for factor screening. For the increases from eight factors to ten factors and from ten factors to twelve factors, the increase in AUC values of the factors screened by the four methods was small. The AUC values for 12 factors obtained with the four screening methods were all between 0.9025 and 0.9103 and were lower those in the case of 20 factors (AUC value is 0.9107).

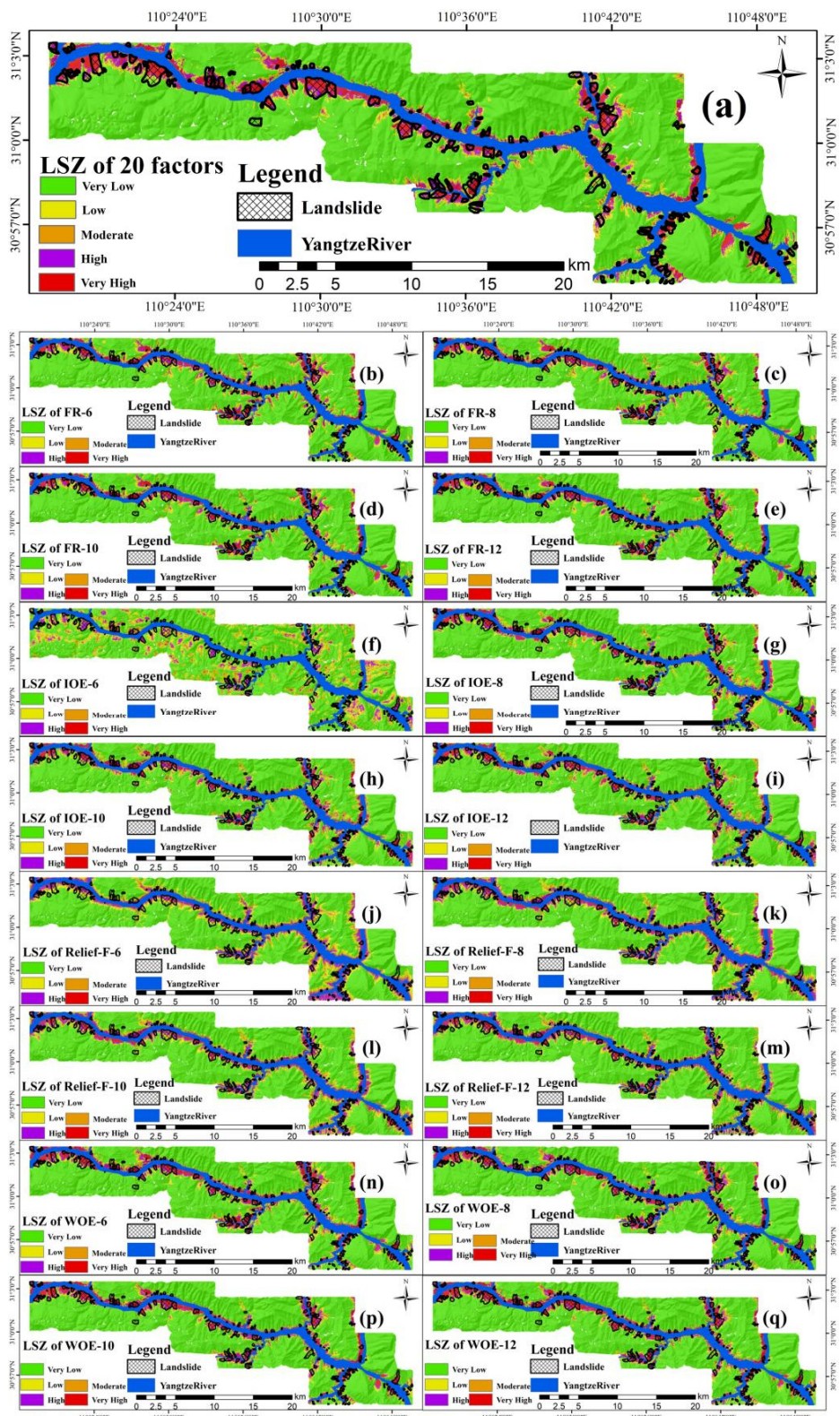

**Figure 7.** LSZ obtained with (**a**) 20 factors, (**b**) 6 factors based on FR, (**c**) 8 factors based on FR, (**d**) 10 factors based on FR, (**e**) 12 factors based on FR, (**f**) 6 factors based on IOE, (**g**) 8 factors based on IOE, (**h**) 10 factors based on IOE, (**i**) 12 factors based on IOE, (**j**) 6 factors based on Relief-F, (**k**) 8 factors based on Relief-F, (**l**) 10 factors based on Relief-F, (**m**) 12 factors based on Relief-F, (**n**) 6 factors based on the WOE Bayesian model, (**o**) 8 factors based on the WOE Bayesian model, (**p**) 10 factors based on the WOE Bayesian model, and (**q**) 12 factors based on the WOE Bayesian model.

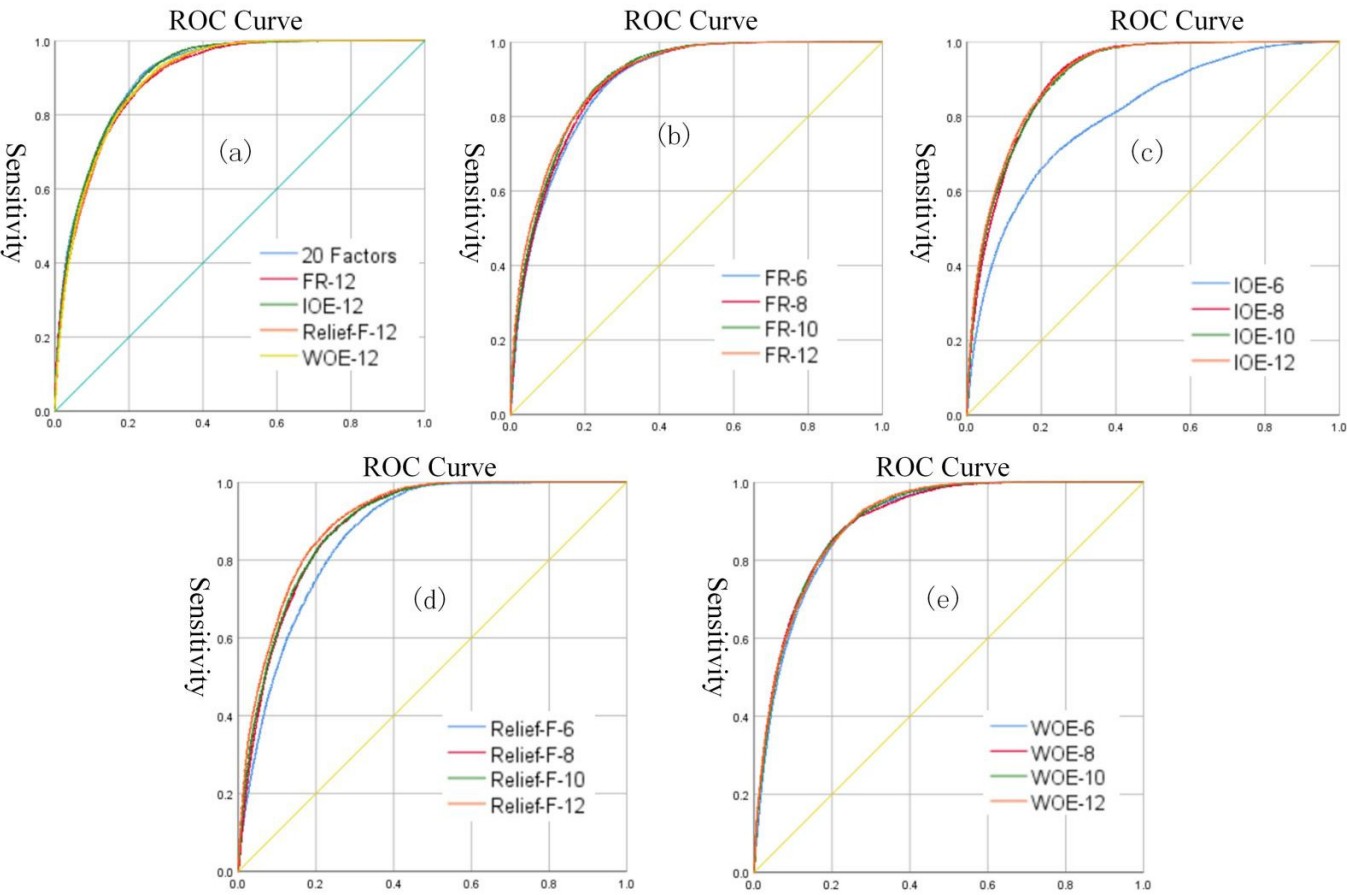

**Figure 8.** ROC curve analysis: (**a**) ROCs of different methods, (**b**) ROC of FR, (**c**) ROC of IOE, (**d**) ROC of Relief-F, (**e**) ROC of the WOE Bayesian model.

### 4.5.2. Specific Category Precision Analysis

The LSM results of the specific category precision analysis based on the four factor-screening methods are shown in Figure 9.

Based on Figure 9, IOE-12 yields the highest specific category precision in the predicted very-high areas of landslides, with a value of 27.06%. The second highest value is observed for IOE-10 at 26.76%. The specific category precision obtained with 20 factors is only the fourth highest, with a value of 26.61%. WOE-12, which ranked fourth in terms of AUC value in Section 4.5.1, ranked only eleventh in specific category precision, with a value of 23.14%.

### 4.5.3. Evaluation of Statistical Measures

The results of the calculation of five statistical measures, namely OA, precision, recall, the F-measure, and the MCC, are shown in Figure 10.

Figure 10 shows that as the number of factors increases, the OA, precision, F-measure, and MCC all increase. A comparison indicates that in addition to the recall, the OA, precision, F-measure, and MCC of the 20-factor case were the highest. The WOE-12 and WOE-10 scenarios yield the closest values of the five statistical measures to those in the 20-factor case. IOE-6 yields the lowest values of all five statistical measures. The lowest recall rate was associated with IOE-6, with a value of 0.7577, and the second lowest recall rate was for the case with 20 factors, with a value of 0.7938.

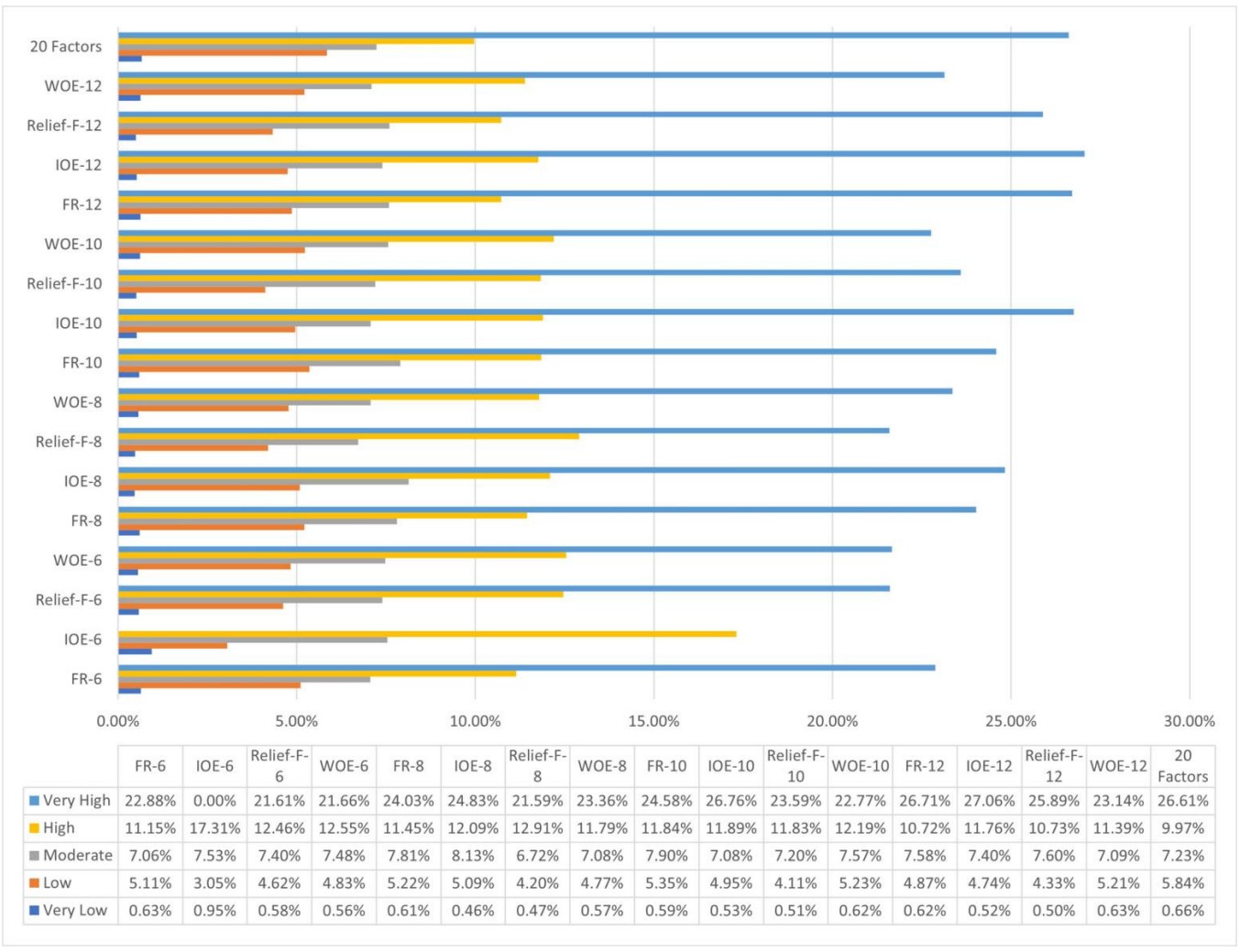

**Figure 9.** Specific category precision analysis.

In summary, Figure 8 shows that except the ROC curve for the 20-factor, IOE-12 performs well, and the ROC curves are nearly coincident with that of the 20-factor. Table 7 indicates that the AUC value for 20-factor is 0.9107, and the AUC value for IOE-12 is 0.9103, which is similar. The highest specific category precision in a very-high area in Figure 9 is observed for IOE-12, with a value of 27.06%, and the specific category precision for the 20-factor case is 26.61% for these areas, ranking only third. The ROC curves, AUC values, and the specific category precision results of very-high landslide areas suggest that the accuracy is similar for the LSM of IOE-12 and the 20-factor, yielding accurate LSI results. The 20-factor, IOE-12, and IOE-10 cases all yield reasonable AUC and specific category precision values of very-high landslide areas, but all five statistical measures are not good in each case. Figure 10 shows that the OA in the 20-factor case is 84.19%, precision is 0.1207, recall is 0.7938, the F-measure is 0.2095, and the MCC is 0.2970; the OA, precision, F-measure, and MCC are the highest, but recall is relatively low, ranking second from the bottom in the 17 groups of experiments. The reason for the high OA, precision, F-measure, and MCC values in the 20-factor case is that FP is small and FN is large in the confusion matrix, which indicates that in the 20-factor landslide prediction result, the raster units that were originally non-landslides are commonly predicted as landslides, but few actual landslides were predicted to be non-landslides. In Figure 10, the OA, precision, recall, F-measure, and MCC results are all good for IOE-12, with values of 81.16%, 0.1077, 0.8423, 0.1910, and 0.2823, respectively. Among the five statistical measures in the 17 groups of experiments, the OA of IOE-12 ranked second, and the precision, recall, F-measure, and

MCC all ranked sixth; the difference between the five calculated indicator values and the highest values of these indicators in the 17 groups of experiments was small. Thus, the overall prediction effect was the best for IOE-12, and this approach considers safety and cost. In summary, the importance of the LSM factors obtained by using different screening methods was different; the IOE-12 and 20-factor cases yielded similar ROC curve, specific category precision, and statistical measure results; as the degree of screening decreased, the LSM results obtained when more factors were retained improved; additionally, when no fewer than eight factors were retained, the IOE method outperformed FR, Relief-F, and WOE Bayesian modeling methods.

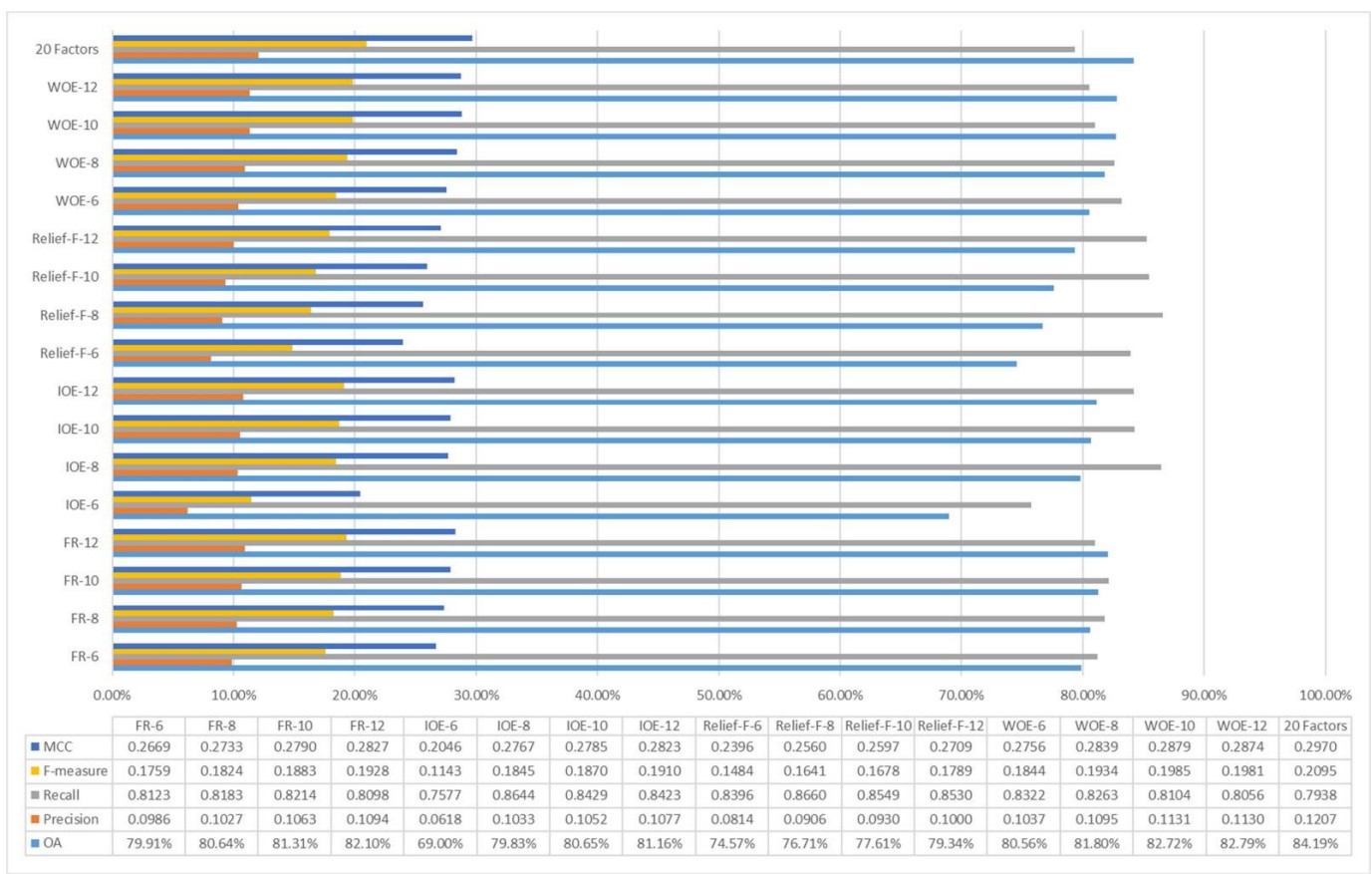

**Figure 10.** The results of five statistical measures.

## 5. Discussion

### 5.1. Quantitative Analysis of the LSM Evaluation Results

To facilitate a comparison of the advantages and disadvantages of several methods, the scoring method is used to comprehensively quantify the ROC curves, specific category precision analysis, and five statistical measure results in LSM. The scoring rules are as follows: the number of models in this study is 17, so the highest possible score is 17 points, and the lowest is 1 point. The statistical method is based on the average of the five algorithms. The scores are shown in Table 8.

**Table 8.** Score table of LSM results' evaluation for important subset of factors.

| Number of Factors | Model | AUC | Specific Category Precision Analysis-Very High | Five Statistical Measures | | | | | | Total Score |
|---|---|---|---|---|---|---|---|---|---|---|
| | | | | OA | Precision | Recall | F-Measure | MCC | Average Score | |
| 6 factors | FR + SVM | 3 | 6 | 7 | 5 | 6 | 5 | 5 | 5.6 | 14.6 |
| | IOE + SVM | 1 | 1 | 1 | 1 | 1 | 1 | 1 | 1 | 3 |
| | Relief-F + SVM | 2 | 3 | 2 | 2 | 11 | 2 | 2 | 3.8 | 8.8 |
| | WOE + SVM | 7 | 4 | 8 | 9 | 10 | 8 | 8 | 8.6 | 19.6 |
| 8 factors | FR + SVM | 6 | 10 | 9 | 7 | 7 | 7 | 7 | 7.4 | 23.4 |
| | IOE + SVM | 14 | 12 | 6 | 8 | 16 | 9 | 9 | 9.6 | 35.6 |
| | Relief-F + SVM | 4 | 2 | 3 | 3 | 17 | 3 | 3 | 5.8 | 11.8 |
| | WOE + SVM | 9 | 8 | 13 | 14 | 9 | 14 | 14 | 12.8 | 29.8 |
| 10 factors | FR + SVM | 8 | 11 | 12 | 11 | 8 | 11 | 11 | 10.6 | 29.6 |
| | IOE + SVM | 15 | 16 | 10 | 10 | 13 | 10 | 10 | 10.6 | 41.6 |
| | Relief-F + SVM | 5 | 9 | 4 | 4 | 15 | 4 | 4 | 6.2 | 20.2 |
| | WOE + SVM | 10 | 5 | 15 | 16 | 5 | 16 | 16 | 13.6 | 28.6 |
| 12 factors | FR + SVM | 12 | 15 | 14 | 13 | 4 | 13 | 13 | 11.4 | 38.4 |
| | IOE + SVM | 16 | 17 | 11 | 12 | 12 | 12 | 12 | 11.8 | 44.8 |
| | Relief-F + SVM | 11 | 13 | 5 | 6 | 14 | 6 | 6 | 7.4 | 31.4 |
| | WOE + SVM | 13 | 7 | 16 | 15 | 3 | 15 | 15 | 12.8 | 32.8 |
| 20 factors | SVM | 17 | 14 | 17 | 17 | 2 | 17 | 17 | 14 | 45 |

Based on Table 8, when the number of factors increases, the score will increase, indicating that when the degree of factor screening is low, retaining more factors will improve the effectiveness of LSM to a certain extent; however, the degree of improvement achieved with different methods varies. In this study, when the number of the screening factors retained was greater than 10, IOE > FR > WOE Bayesian model > Relief-F. When the number of the screening factors retained was eight, IOE > WOE Bayesian model > FR > Relief-F. When the number of the screening factors retained was six, the WOE Bayesian model > FR > Relief-F > IOE. Therefore, the following conclusions can be drawn: (1) Relief-F is relatively ineffective at different levels of screening; (2) When the degree of screening is high, the IOE method is not appropriate, and key factors are missed; (3) When screening for eight or more retained factors, IOE performs best among the four methods; (4) When the screening degree is high, the effect of the WOE Bayesian model is better than that of FR, and when the degree of screening is relatively low, the effect of FR is better than that of the WOE Bayesian model; (5) The 20-factor LSM approach scored 45 points, while IOE-12 scored 44.8 points, a small difference; therefore, the IOE method was used in this study to screen 12 retained factors (TWI, TPI, slope, terrain surface texture, the convergence index, convexity, elevation, fault, mid-slope position, slope structure, MRN, and aspect) and can ensure the accuracy of landslide-prone results, and the remaining eight factors (flow width, slope height, valley depth, slope length, flow line curvature, lithology, SPI, and total curvature) were considered to have little impact on the occurrence of landslides and are noncritical factors.

Reichenbach counted the number of LSM factors used in a review paper and found that 596 factors were used in landslide studies, but 445 factors were used only once or twice. Of remaining factors, 10.5% were slope factors, 9.2% were geo-lithological factors, 8.1% were aspect factors, and 7.3% were river/catchment factors; additionally, curvature factors accounted for 7.2%, other morphometric factors accounted for 5.6%, elevation factors accounted for 5.2%, soil factors accounted for 5.2%, distance to fault accounted for 3.5%, and geo-structural factors accounted for 3.4% [20]. The 12 factors retained in IOE screening are all commonly used and important factors in landslide studies.

*5.2. Reordering the Retained LSM Factors Based on the Increase in Scores and the Increase in Related Factors*

By combining Tables 6 and 8, it can be found that the importance of the factors obtained by screening with different methods varies, and as the number of factors increases, the AUC value, the specific category precision, and the five statistical measures all increase, but to different degrees. Table 8 indicates that the increase in score when the number of factors was increased was large, suggesting that these increased factors were important for

the landslide occurrence prediction. To facilitate a comparison of the increases in score for different numbers of factors, Figure 11 was created.

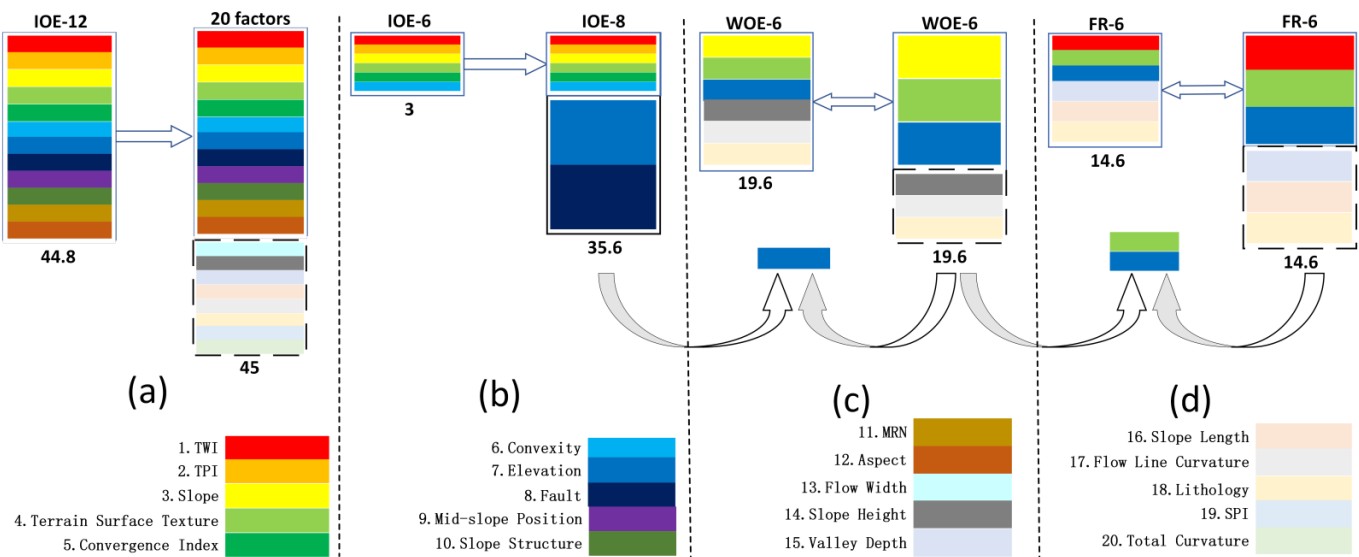

**Figure 11.** Comparison of the increases in scores and in related factors: (**a**) Comparison of the score of the LSM evaluation of IOE-12 and the score of the LSM evaluation of 20 factors; (**b**) Comparison of the score of the LSM evaluation of IOE-6 and the score of the LSM evaluation of IOE-8; (**c**) The score of the LSM evaluation of WOE-0; (**d**) The score of the LSM evaluation of FR-6.

As shown in Figure 11a, the score in the 20-factor case is 45 points, and the IOE-12 score is 44.8 points, indicating that eight factors (flow width, slope height, valley depth, slope length, flow line curvature, lithology, SPI, and total curvature) have little effect on landslide occurrence prediction and are noncritical factors; thus, they can be removed directly from the analysis. As shown in Figure 11b, the highest increase in score is obtained when the number of IOE screening factors is increased from six to eight, and the score is increased by 32.6 points, indicating that considering the elevation and fault factors leads to an increase in the score. As shown in Figure 11c, the second highest score increase is associated with the use of the WOE Bayesian model and six screening factors, with a score increase of 19.6 points. After removing the unimportant factors, three factors were retained (elevation, slope, and terrain surface texture); thus, the score increase was caused by at least one of these 3 factors. As shown in Figure 11b,c, the two factor sets with the highest increased scores both included elevation, so the addition of elevation may have led to a significant increase in scores. Similarly, in Figure 11c,d, the two factor sets with the highest increased scores both included elevation and terrain surface texture, so the addition of these factors may have led to a significant increase in scores. By analogy, the remaining factors can also be assessed from multiple sets of comparisons: if an increase in the factor occurs multiple times in multiple groups with a larger increase in score, it indicates that the factor is responsible for the increase in score. Ranked based on the highest to lowest score increases, the importance of factors is as follows: elevation, terrain surface texture, slope, TWI, convexity, slope structure, mid-slope position, convergence index, fault, aspect, MRN, and TPI. To verify the correctness of these results, six core factors, eight core factors, and ten core factors were selected according to the importance of factors obtained in the new ranking, and after performing LSM using the SVM model, the results were evaluated based on the ROC curve, specific category precision. and five statistical measures. The evaluation results for the three groups of experiments were scored, and the results are shown in Table 9.

**Table 9.** Evaluation of the LSM results for different core factor sets.

| Factors | AUC | Specific Category Precision—Very High | Five Statistical Measures | | | | | Total Score |
|---|---|---|---|---|---|---|---|---|
| | | | OA | Precision | Recall | F-Measure | MCC | |
| 6 factors | 0.9000 | 24.86% | 79.32% | 0.0996 | 0.8503 | 0.1783 | 0.2703 | 27.55 |
| 8 factors | 0.9062 | 26.84% | 80.27% | 0.1039 | 0.8493 | 0.1852 | 0.2769 | 41.72 |
| 10 factors | 0.9082 | 26.83% | 80.93% | 0.1063 | 0.8405 | 0.1888 | 0.2801 | 42.88 |

A comprehensive analysis of Tables 8 and 9 shows that the score for all 10 core factors is higher than the scores for the 10 important factors individually. The score for the eight core factors is much higher than the scores for the eight important factors individually and higher than those for FR-12, WOE-12, and Relief-F-12. The score for the six core factors was higher than that for the combination of all six important factors and those for FR-8 and Relief-F-10.

A significant increase in the score after reordering the importance of the factors based on the score increase indicates that the previous assumption is correct regarding factors present in multiple groups with large score increases being the most important. Therefore, among the FR, IOE, Relief-F, and the WOE Bayesian modeling methods, IOE is the best of the four screening methods when the retained number of screening factors is eight or greater. In addition, the importance of the factors was sorted by using the FR, IOE, Relief-F, and WOE Bayesian modeling methods considering different degrees of screening, and the importance of the factors was reranked according to the score increase and a comprehensive comparison, which approach improved the accuracy of the importance ranking of factors and increased the effectiveness of factor screening.

## 6. Conclusions

In this paper, with the Three Gorges Reservoir area to the Padang section as an example, 40 factors were extracted from the collected geological, topographical, hydrological, remote sensing, and other data. After PCC was used to remove factors with correlation greater than 0.6 and VIF was used to remove factors with multicollinearity greater than 10 and less than 0.1, 20 LSM factors were obtained. FR, IOE, Relief-F, and WOE Bayesian modeling methods were used to sort the importance of the 20 LSM factors. To study the influence of the degree of screening of these four methods on LSM, different sets of important factors (six factors, eight factors, ten factors, and twelve factors) were screened according to the importance of the factors for comparison. Then, the same training sample validation sample of screened factors was used by an SVM to generate LSM results, and the LSM results generated with different factor combinations were evaluated based on ROC curves, specific category precision analysis, and five statistical measures. To facilitate comparison, the evaluation results were scored by comprehensive quantitative analysis. By scoring the LSM results generated with the important factor sets obtained using the four screening methods with different screening degrees, it was found that when simplified to eight factors, ten factors, and twelve factors, the effect of IOE screening was the best; however, the effect of IOE screening was very poor when 6 factors were retained. In addition, the screening effect of Relief-F was worse than that of the FR and WOE Bayesian models in most cases. The evaluation score of the LSM result with IOE screening for 12 factors was 44.8 points, and that in the 20-factor case in LSM was 45 points, a small difference. Thus, it was speculated that eight factors (flow width, slope height, valley depth, slope length, flow line curvature, lithology, SPI, and total curvature) have little effect on the LSM result and are noncritical factors; thus, they were screened and removed. In the discussion, the four screening methods were comprehensively compared, and the importance of the remaining 12 factors was sorted by comparing the score increases of the four screening methods and different screening degrees; from largest to smallest, the most important factors were as follows: elevation, terrain surface texture, slope, TWI, convexity, slope structure, mid-slope position, convergence index, fault, aspect, MRN, and TPI. According

to these twelve reordered factors, the six, eight, and ten most important core factors were selected, and SVM modeling was performed. The ROC curves, specific category precision analysis results, and five statistical measures were used to evaluate the LSM results, and it was found that the core factor approaches improved the four screening methods. Not only were the scores improved, but the accuracy of the importance ranking of factors also increased after reordering. Therefore, by using the FR, IOE, Relief-F, and WOE Bayesian models to screen the factors, comprehensively comparing the score increases achieved with the four methods and the factors that led to the increases and reordering the importance of the factors, the accuracy of the importance scores and rankings of factors can be improved.

**Author Contributions:** Writing-original draft preparation, T.X. and X.Y.; conceptualization, T.X. and X.Y.; writing-review and editing, T.X. and X.Y.; validation, X.Y. and J.Z.; visualization, X.Y., W.J. and J.Z.; formal analysis, T.X. and J.Z.; investigation, X.Y. and J.Z.; resources, T.X. and W.J.; data curation, W.J. and J.Z.; methodology, T.X., X.Y. and W.J.; supervision, T.X.; project administration, T.X.; software, X.Y. and J.Z.; funding acquisition, T.X. and W.J. All authors have read and agreed to the published version of the manuscript.

**Funding:** This study was funded by the National Natural Science Foundation of China (No. 41807297), National Natural Science Foundation of China (No. 42101375) and Innovation Demonstration Base of Ecological Environment Geotechnical and Ecological Restoration of Rivers and Lakes (No. 2020EJB004).

**Institutional Review Board Statement:** Not applicable.

**Informed Consent Statement:** Not applicable.

**Data Availability Statement:** Remote sensing data and DEM data can be downloaded from public websites. However, basic geographic data, basic geological data, and landslide distribution data are all confidential data in China. According to relevant regulations, these confidential data have been decrypted when we use them. Any researchers in related fields that need these decrypted data can contact the corresponding author.

**Acknowledgments:** We are grateful to the National Natural Science Foundation of China and Innovation Demonstration Base of Ecological Environment Geotechnical and Ecological Restoration of Rivers and Lakes. We are also thankful to the Headquarters of Prevention and Control of Geo-Hazards in the Area of the Three Gorges Reservoir for providing data and material. We thank the anonymous reviewers for their constructive comments and suggestions on the manuscript. We also thank editorial employees for editing the manuscript.

**Conflicts of Interest:** The authors declare no competing interest.

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
