# Peer review of "Comparative Assessment of the Efficacy of the Five Kinds of Models in Landslide Susceptibility Map for Factor Screening: A Case Study at Zigui-Badong in the Three Gorges Reservoir Area, China"

_sustainability, doi:10.3390/su15010800_

Round 1

Reviewer 1 Report

General comments:

1.      The manuscript follows the MDPI template (figures, references etc) with some minor exceptions

2.      The abstract provides sufficient information about the manuscript content

3.      The use of the English needs a revision.

4.      Most of the figures are appropriate and well presented

For specific comments, please refer in the manuscript.

Overall, my suggestion is that the paper should be accepted after minor improvements

Author Response

Reviewer1:

  • The manuscript follows the MDPI template (figures, references etc) with some minor exceptions:

R: Thank for your advice.

We have checked and modified all the figures and references according to MDPI template.

We hope you are satisfied with this reply.

  • A brief introduction of the data processing platform in Table 2 is enough, without table explanation;

R: Thank for your advice.

At your suggestion, we have removed the table explanation “The prediction results are divided into four situations: true positive (TP), false negative (FN), false positive (FP), and true negative (TN).” in Ln 254 ~ 255.

We hope you are satisfied with this reply.

3) The use of the English needs a revision.

R: Thank for your advice.

  We tried our best to translate the article into English and proofread it sentence by sentence. In addition, we invited AJE to polish our English manuscript.

We hope you are satisfied with this reply.

4) Most of the figures are appropriate and well presented

R: Thank for your advice.

It's a great honor to receive your praise.

Reviewer 2 Report

Excessive landslide susceptibility mapping (LSM) factors can reduce the accuracy of LSM results, and are not conducive to researchers finding the key LSM factors. Author have used different method to screen evaluation factors, and different LSM results have been obtained. LSM results for different numbers of factors, thus verifying the effectiveness of the proposed method for ranking the importance of LSM factors. The method proposed in this study can effectively screen the key LSM factors and improve the accuracy and scientific soundness of LSM results. This study was very meaning, overall, I recommend minor revise following the comments below.

1.      Ln 45 ~ 53, different LSM method have been listed, you should be summarized advantages and disadvantages of each method.

2.      Ln 71, the relevantly evidence has to supplement, and you should be further explained.

3.      Ln 72 ~ 96, the statistical methods achievements have been presented, others’ achievements (machine learning method)?

4.      I suggest that ‘figure 1’ move to method section.

5.      Ln 135, ‘km2’ change to ‘km2’.

6.      I suggest that ‘Yellow River’ and ‘Beijing’ are added in the figure, and the figure was spilt, meanwhile, marked a, b and c.

7.      Ln 146 ~ 162, content should be simplified.

8.      Section of ‘introduction to the research methods’ should be simplified.

9.      I suggest that section of ‘4.1 data and samples’ move to section of ‘overview of the study area and data introduction’.

10.  Section of ‘4.2 data preprocessing’ move to section of ‘method’.

11.  The legend of figure was too small in figure 3.

12.  I suggest that figure 4 was used as a supplement material.

13.  The legend of figure was too small in figure 5. In addition, the spatial distribution of landslide susceptibility should be added.

Reviewer 3 Report

Authors report the comparative assessment of the efficacy of the five kinds of models in landslide susceptibility map for factor screening: a case study, the result may provide the basis for increasing the understanding of the behavior of landslide safety. The whole manuscript needs minor revisions before publication.
The manuscript is recommended publication after the following revisions are made.
(1) The title of the manuscript should be “Comparative assessment of the efficacy of the five kinds of models in landslide susceptibility map for factor screening: a case study”.
(2)  More relative references should be cited and reviewed, and the contribution of the manuscript should be explicitly pointed out. 
(3)  The most important, at least a sustainable pattern of landslide safety management from the Three Gorges Reservoir area  should be added.(4) Although the reviewer is not English native, I recommend the whole manuscript language improvement.

Author Response

Reviewer3:

  • The title of the manuscript should be “Comparative assessment of the efficacy of the five kinds of models in landslide susceptibility map for factor screening: a case study”.

R: Thank for your advice.

At your suggestion, we’ve changed the title to “Comparative assessment of the efficacy of the five kinds of models in landslide susceptibility map for factor screening: A case study at Zigui-Badong in the Three Gorges Reservoir Area, China”.

We hope you are satisfied with this reply.

  • More relative references should be cited and reviewed, and the contribution of the manuscript should be explicitly pointed out.

R: Thank for your advice.

68 references were cited in the original manuscript and 72 in the revised manuscript. Among these literatures, several are review literatures. All the references are from recent years, and relevant references have described the views cited in this paper. For example: “The commonly used statistical methods are the certainty factor (CF) [32-35], frequency ratio (FR) [36-39], index of entropy (IOE) [40,41], weight of evidence (WOE) [36,40,42,43] and relief algorithm methods [44-47] et al. Wu et al. used rough set and correlation coefficient analysis to screen 12 key environmental factors from 22 overall landslide factors for LSM [1]. Dou et al. used the CF method to optimize 14 possible LSM factors and selected slope angle, aspect, diversion density network, distance from a geological boundary, distance from a fault, and lithology as factors for further analysis [48].

We hope you are satisfied with this reply.

  • The most important, at least a sustainable pattern of landslide safety management from the Three Gorges Reservoir area should be added.

R: Thank for your advice.

In the future, we will do further research on the governance of the Three Gorges Reservoir area, including the sustainable pattern of landslide safety management and landslide cost management. At present, this paper focuses on the early factor selection stage of landslide susceptibility evaluation. Due to the workload, it is not possible to cover all the areas of landslide, but we will gradually study other aspects of landslide in the future.

We hope you are satisfied with this reply.

  • Although the reviewer is not English native, I recommend the whole manuscript language improvement.

R: Thank for your advice.

We tried our best to translate the article into English and proofread it sentence by sentence. In addition, we invited AJE to polish our English manuscript.

We hope you are satisfied with this reply.

Reviewer 4 Report

In this study authors give an interesting research, using the Three Gorges Reservoir area to the Padang section as an example, the frequency ratio (FR), index of entropy (IOE), relief-F algorithm and weights-of-evidence (WOE) Bayesian model were used to sort and screen the importance of 20 LSM factors; then, a support vector machine (SVM) model was established based on different factor sets to generate an LSM, and the LSM results were evaluated by using the receiver operating characteristic (ROC) curve, specific category precision analysis and five statistical measures; finally, the results of the above three evaluation methods were comprehensively quantitatively scored. The results showed that the IOE screening factor was better than the FR, Relief-F and WOE Bayesian models in the case of retaining no less than 8 factors; the score for 20 factors without screening was 45 points, and the score for 12 factors screened based on the IOE was 44.8 points, indicating that there was an optimal retention number that had little effect on the LSM results when IOE screening was used. Based on the information provided in the methods section, the findings of the manuscript are replicable. The writing in the text is excellent, and readers should find it to be quite interesting. All charts' figures, might be larger. 

Author Response

Reviewer4:

  • All charts' figures, might be larger. 

R: Thank for your advice.

We have increased the resolution of all the charts in order to make the charts' figures larger and easier to see.

We hope you are satisfied with this reply.

Reviewer 5 Report

This research focused on exploring a series of methods to sort and screen the importance 20 landslide susceptibility mapping (LSM) factors. These methods include frequency ratio (FR), index of entropy (IOE), relief-F algorithm, weights-of-evidence (WOE) Bayesian model, and support vector machine (SVM). The LSM results were evaluated by the receiver operating characteristics (ROC) curve, specific category precision analysis, and five statistical metrics. The reviewer believes that the current version of the manuscript is not yet ready for publication; the authors are encouraged to consider the following comments and suggestions and revise the manuscript accordingly.

1. The authors should consider streamlining the Abstract section to show more context about this research. Currently, it does not provide an overview of the research.

2. The authors should split Introduction section into two sections, including Introduction and Related Work section. The Introduction section should focus on introducing the research objectives and research questions to answer, while the Related Work section should focus on literature review of related work or previous work and defining the research gap. In addition, the authors used remote sensing imagery but did not provide further discussion on this type of data. The authors should discuss if unmanned aircraft systems (UAS) collected remote sensing imagery could be used to improve the proposed method. The authors should read and cite the paper of “The Impact of Small Unmanned Airborne Platforms on Passive Optical Remote Sensing: A Conceptual Perspective.”

3. The authors should provide more information about the remote sensing data that were used in this study. The only information that the reviewer can find is that the remote sensing data have a spatial resolution of 30 meters. By the way, plural format should be used for data. The authors should discuss where find the remote sensing data, how they were processed, and what products were derived from these data. Clearly, DEM data were used, but the authors also stated that the remote sensing data is 30m. Are you referring to the same dataset?

4. The authors should use more generally accepted terms in remote sensing or photogrammetry for their manuscript. For example, the authors kept using high resolution to describe the satellite images. What resolution do you refer to for this statement? There are four resolutions for aerial imagery, including spatial resolution, spectral resolution, temporal resolution, and radiometric resolution. I think the authors are indicating spatial resolution, but it needs to be clearly stated in the manuscript.

5. More detailed information should be provided regarding the SVM. The authors should provide a document to show all the parameters that were used for SVM and each image-processing step. This document can help researchers who want to replicate the research.

6. All figures are blurry and the authors should improve them. The authors should consider converting their figures to vector graphics to improve the readability.

Round 2

Reviewer 1 Report

Dear Editor,

I would like to thank you for giving me the opportunity to re-review this interesting research by Yu Xianyu, Xiong Tingting, Jiang Weiwei and Zhou Jianguo. I have carefully read the paper and with the changes made by the authors, I believe that it can be accepted at present form.